# Extent of gross underestimation of precipitation in India

Gopi Goteti[1] and James Famiglietti[2]

[1]5741 NW 92nd Ct, Johnston, Iowa, 50131, USA
[2]School of Sustainability, College of Global Futures, Arizona State University, Tempe, AZ, USA

**Correspondence:** Gopi Goteti (saagu.neeru@gmail.com)

**Abstract.**

Underestimation of precipitation ($UoP$) in the hilly and mountainous parts of South Asia is estimated by some studies to be as large as the observed precipitation ($P$). For instance, correction factors (CFs) developed by the recent PBCOR dataset have values exceeding 2.0 across the wettest regions of India, some of which have experienced catastrophic flooding in the recent 5 past. However, $UoP$ has been analyzed only to a limited extent across India. Towards bridging this gap, this study analyzes watershed-scale $UoP$ using various $P$ datasets within a water *im*balance analysis. Among these $P$ datasets, the often-used Indian Meteorological Department (IMD) dataset is of primary interest.

Gross $UoP$ was identified by analyzing the extent of imbalance in the annual water budget of watersheds corresponding to 242 river gauging stations where quality controlled data on catchment boundaries and streamflow is available. Water year 10 (WY) based volume of observed annual $P$ was compared against observed annual streamflow ($R$) and satellite-based actual evapotranspiration ($ET$). Across many watersheds of both Northern and Peninsular India, the spurious water imbalance scenarios of $P \leq R$, or $P << R + ET$, were realized. It is shown that management of water, such as groundwater extraction, reservoir storage and water diversion (imports or exports), is generally minimal compared to annual $P$ in such watersheds. It is also shown that annual changes in terrestrial water storage are minimal compared to annual $P$ in such watersheds. Assuming 15 data on $R$ (and $ET$ to a lesser extent) to be reliable, it is concluded that $UoP$ is very likely the cause of such imbalance. Groundwater flow across topographic boundaries, or inter-watershed groundwater flow (IGF), is assumed to be negligible in this study. While the effect of IGF on $R$ is unknown, examples are provided to show that IGF is unlikely the cause of observed imbalance in certain watersheds.

All 12 of the $P$ datasets analyzed here suffer from $UoP$, but the extent of $UoP$ varies by dataset and region. The reanalysis-20 based datasets ERA5-Land and IMDAA are less affected by $UoP$ than IMD, and the spatial patterns of estimated CFs based on these two datasets are also consistent with those made independently by the PBCOR dataset. Based on the 30-year period of WY 1985-2014, $P$ for the whole of India could be up to 19% (ERA5-Land) to 37% (IMDAA) higher than IMD, with substantial variability within years and river basins. For instance, $P$ for the Indian portion of the Ganga River Basin, for the same 30-year period, could be up to 36% (ERA5-Land) to 54% (IMDAA) higher than IMD. The actual magnitude of $UoP$ is 25 speculated to be even greater. Moreover, trends in IMD's $P$ are not always present in ERA5-Land and IMDAA. Studies using IMD should exercise caution since $UoP$ could lead to misrepresentation of water budgets and long-term trends.

The empirical approach of identifying watersheds affected by $UoP$ using a water imbalance approach is contingent on data availability. It is speculated that if additional data on $R$ becomes available, particularly in Northern India, many other watersheds affected by $UoP$ would be identified. While the scientific community is striving to continually improve $P$ products, India's water agencies can help the community better quantify $UoP$ by making observed hydrometeorological data more widely available. Limitations of this study are discussed.

## 1  Introduction

Precipitation ($P$) is a key component of the hydrological cycle, and changes in spatial and temporal patterns of precipitation due to climatic change is a very important area of concern (Krishnan et al., 2020). Such changes are particularly relevant for India where a substantial portion of its population relies on an agrarian economy, which in turn is strongly tied to specific seasonal patterns of precipitation (Chauhan et al., 2014). Thus, accurate measurement of precipitation and subsequent dissemination of such measurements is important for socioeconomic purposes. Raw data from rain gauges is often compiled by government or research agencies to create precipitation products for subsequent use in hydrological and other environmental studies. Other precipitation products based on satellites, reanalysis, weather simulators, or a combination of the above sources are also available (Sun et al., 2018). Several studies have analyzed such products across the whole of India - e.g., Rana et al. (2015), Prakash (2019), and Gupta et al. (2020), and specific regions of India - e.g., Thakur et al. (2019) and Kanda et al. (2020). Within these studies, gauge-based precipitation products are often treated as reference products, or benchmarks, when evaluating satellite-based and other non-traditional datasets.

In hydrological and meteorological studies across India, the de facto benchmark dataset is the gauge-based gridded daily product from the Indian Meteorological Department (IMD) (Pai et al., 2014). Since its latest release in 2014, the IMD dataset has been cited more than 1,300 times on Google Scholar alone (https://scholar.google.com/, as of April 23, 2024). However, gauge-based gridded datasets can suffer from inadequate representation of extreme events - such as those reported by King et al. (2013) in Australia; spurious trends due to changes in the locations of reporting gauges - such as those reported by Lin and Huybers (2019) using the IMD dataset; or uncertainties introduced by the relative positioning of reporting gauges - such as those reported by Prakash et al. (2019) using the IMD dataset. Moreover, measurement errors associated with gauges, such as wind-induced undercatch (Adam and Lettenmaier, 2003; Kochendorfer et al., 2017), affect the gridded products which utilize observations from such gauges. Underestimation of precipitation ($UoP$) has been reported in South Asia - e.g., in the upper reaches of the Ganga Basin in Nepal (Dangol et al., 2022) and in the upper reaches of the Indus Basin (Dahri et al., 2018). Studies have also discussed $UoP$ by satellite and gauge-based products in the mountainous regions of India (Li et al., 2017). However, $UoP$ across the whole of India has not been thoroughly analyzed in the literature.

### 1.1  Motivation

A previous study by the first author (Goteti, 2023) noted that many watersheds in the mountainous Western Coast of India have observed annual volume of runoff exceeding the observed annual volume of precipitation. CWC-19 (2019) tabulated

similar exceedances, but did not delve into the details (e.g., Annexure R of CWC-19). It is speculated that such watersheds are affected by $UoP$. Some studies have developed bias-correction factors (CFs) to compensate for $UoP$. Such factors are often developed at the grid resolution of a reference precipitation dataset, typically at average monthly or average annual timescales. For instance, Adam et al. (2006) and Beck et al. (2020) developed grid-based CFs utilizing the concept of Budyko curve.

The PBCOR dataset developed by Beck et al. (2020) estimated bias-corrected precipitation climatology corresponding to several reference climatologies (see Appendix A for further information on the PBCOR dataset). The ratio of bias-corrected annual precipitation from PBCOR to that from IMD is shown in Figure 1. It is evident that the largest ratios occur in the wettest regions of India - the Western Coast of India, Northernmost India and Northeastern India. If estimates from PBCOR are reasonable, it would imply that observed precipitation in these regions, and India in general, is substantially underestimated. Some of the wettest regions of India have experienced catastrophic flooding in the recent past (e.g., Hunt and Menon, 2020; Mahto et al., 2023). Thus, unbiased estimates of precipitation are important for flood and other water resources management. Moreover, significant decreasing trends in precipitation across India have been reported, including the wettest parts of India (e.g., Krishnan et al., 2020). It is important to understand to what extent such trends are affected by $UoP$. Identification and quantification of $UoP$ across India is important for many reasons, but has not received much attention from the scientific community. Filling such a void is the motivation behind this study.

The objective of this study is to analyze the spatial extent and magnitude of $UoP$ in India. This study identifies $UoP$ by analyzing the water balance (or lack of it - i.e., imbalance) where reliable hydrometeorological data is available. By eliminating two potential causes of such annual water imbalance - namely large-scale management and substantial changes in annual terrestrial water storage ($\Delta TWS$), this study concludes that the likely cause of such an imbalance is $UoP$. Other potential causes of water imbalance are also discussed. The $UoP$ analyzed here is watershed-scale or gross $UoP$ and not station-scale $UoP$. A station-scale analysis of $UoP$ is beyond the scope of this study because needed station-wise data is unavailable.

The overall methodology of this study and the specific objectives are illustrated in the flowchart in Figure 2. The specific objectives of this study are: (1) analyze the annual water budget of watersheds using IMD as the source of precipitation, and identify the imbalanced watersheds; (2) investigate large-scale management and annual changes in terrestrial water storage ($\Delta TWS$) in such watersheds; attribute the cause of imbalance to $UoP$ if management and $\Delta TWS$ are found to be relatively minimal; and (3) analyze the extent of $UoP$ within other state-of-the-art precipitation products, and compare it against $UoP$ in IMD to identify reasonable alternatives, if any, to IMD. The remainder of this paper is organized as follows. In Section 2 the datasets used in the analysis are described. In Section 3 the methodology used to identify $UoP$ is described. In Section 4 the results from this analysis are presented. Limitations of this study are presented in Section 5, followed by the conclusions of this study.

The reader should note the following conventions used throughout this paper. The words catchment and watershed are used interchangeably for smaller watersheds, while the word basin is reserved only for larger watersheds - e.g., the Indus Basin or the Ganga Basin. A reference time period often used in analyzing hydrological variables is the water year (WY). In this study, a WY is defined as the period starting from June 1 and ending on May 31 of the following year. For example, WY 2020 spans

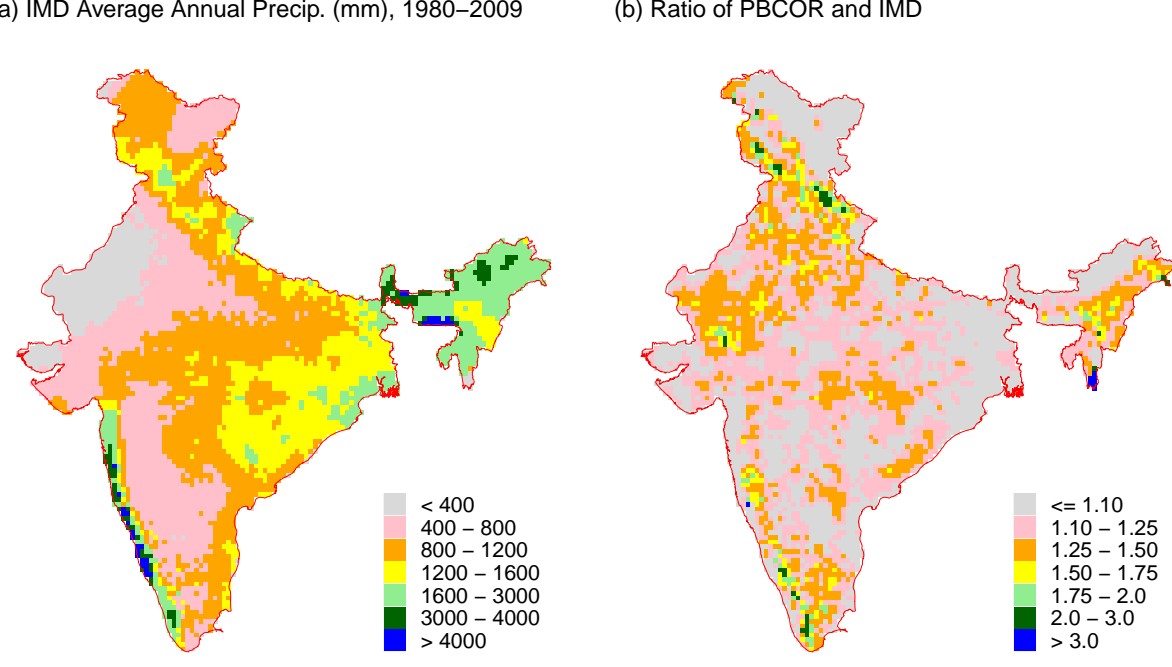

(a) IMD Average Annual Precip. (mm), 1980–2009

(b) Ratio of PBCOR and IMD

| | |
|---|---|
| < 400 | <= 1.10 |
| 400 − 800 | 1.10 − 1.25 |
| 800 − 1200 | 1.25 − 1.50 |
| 1200 − 1600 | 1.50 − 1.75 |
| 1600 − 3000 | 1.75 − 2.0 |
| 3000 − 4000 | 2.0 − 3.0 |
| > 4000 | > 3.0 |

**Figure 1.** (a) Average annual $P$ (mm) from IMD for 1980-2009. (b) Ratio of bias-corrected annual precipitation from PBCOR and annual precipitation from IMD. See Appendix A for additional information on PBCOR.

the period June 1 2020 to May 31 2021. This definition is consistent with the WY definition often used by Indian agencies (e.g., CWC-19, 2019).

## 2 Data

### 2.1 Study domain, river gauging stations and catchment boundaries

The study domain includes the river basins that span India, including the catchment areas that fall outside of the political boundaries of India (Figure 3). The boundaries of the river basins used in this study are generally consistent with those used by India's Central Water Commission (CWC). Consistent with CWC, adjacent watersheds in some regions were pooled to create composite river basins, such as, West Flowing Rivers (WFR) North and South, East Flowing Rivers (EFR) North and South, and West Flowing Rivers of Kutch (WFR Kutch). The catchment boundaries used in this study are from the GHI dataset (Goteti,

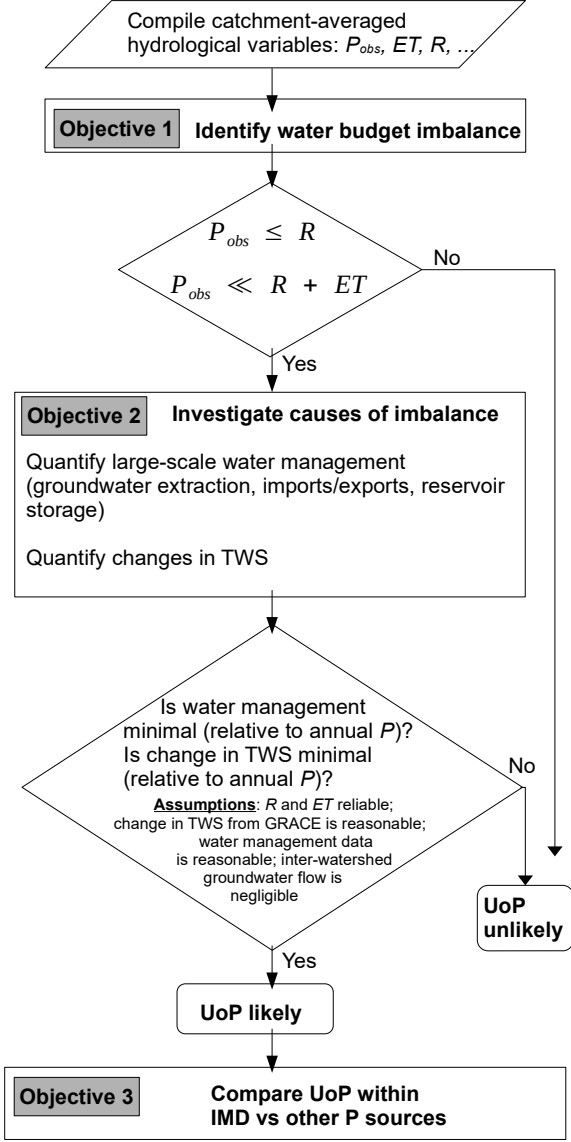

**Figure 2.** Flowchart showing the overall objectives, the methods used to achieve these objectives and the major assumptions made in this study.

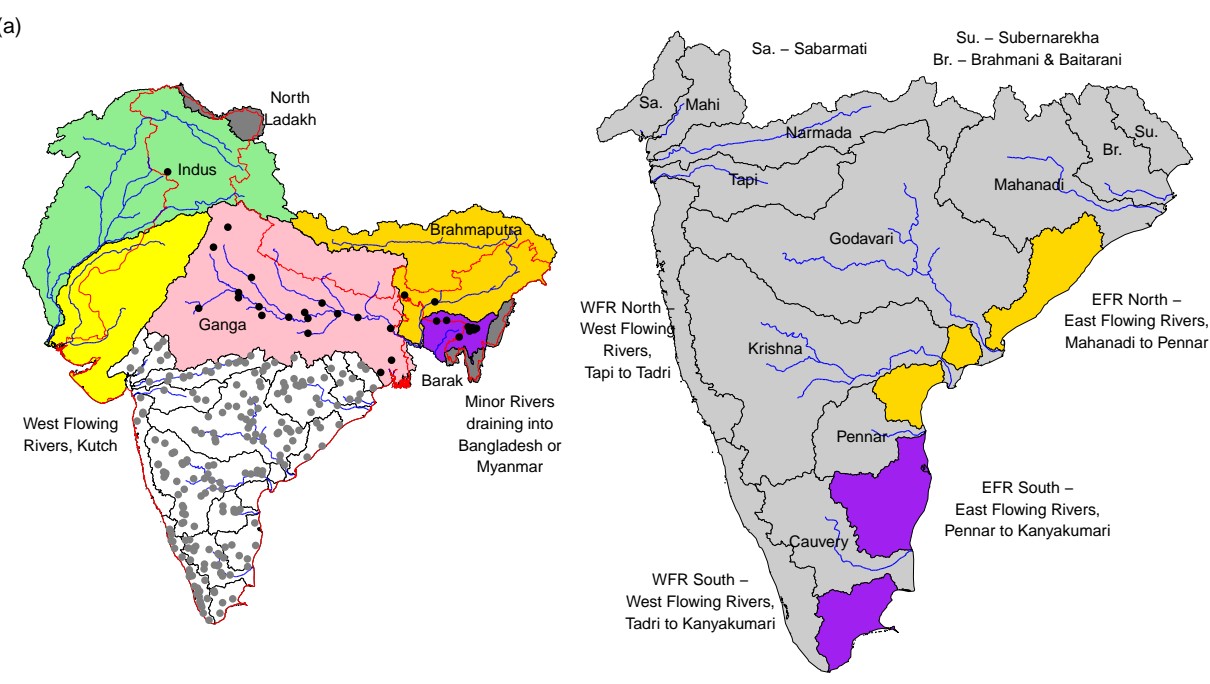

**Figure 3.** (a) Major river basins spanning India. The shaded basins are transboundary basins. Stations with daily streamflow data (grey dots) are from the GHI dataset (Goteti, 2023), and stations with only annual streamflow data (black dots) are from CWC-19 (2019). (b) River basins of Peninsular India. Some basin names are shortened for ease of display within the map, and the complete names are shown next to the map.

2023), a quality-controlled dataset on India's river gauging stations, catchment boundaries and hydrometeorological time series. However, the GHI dataset is limited to Peninsular India. The catchment boundaries for the Northern Indian watersheds were derived using the HydroSHEDS suite of products, using the same procedures as the GHI dataset. Station descriptions available
from CWC were validated using online maps (e.g., Google Maps). Stations were then relocated to the closest point on the river network. The watershed draining into this relocated point, and all of the upstream watersheds were recursively identified using a GIS software. Catchment areas for the delineated watersheds were validated against those reported by CWC.

    The river basins of Peninsular India, non-shaded region in Figure 3 (panel (a)), have daily streamflow data available through India's Central Water Commission (CWC). There is limited streamflow data available for the river basins of Northern India
(shaded regions in Figure 3 (panel (a)). The stations used in this study were chosen such that the catchment area discrepancy between GHI and that published by CWC is less than 5%, and there was at least 5 years of observed streamflow data with

**Table 1.** Summary information on streamflow stations used in this study. Frequency of streamflow observations and the source of the data is also mentioned. Additional information on the individual stations is in the Supplement (Table S6).

| Region | Basin | Stations | Frequency | Source |
|---|---|---|---|---|
| Northern India | Barak | 8 | Annual | CWC-19 |
| Northern India | Brahmaputra | 2 | Annual | CWC-19 |
| Northern India | Ganga | 18 | Annual | CWC-19 |
| Northern India | Indus | 1 | Annual | CWC-19 |
| Peninsular India | Brahmani-Baitarani | 7 | Daily | GHI |
| Peninsular India | Cauvery | 20 | Daily | GHI |
| Peninsular India | EFR North | 5 | Daily | GHI |
| Peninsular India | EFR South | 12 | Daily | GHI |
| Peninsular India | Godavari | 40 | Daily | GHI |
| Peninsular India | Krishna | 42 | Daily | GHI |
| Peninsular India | Mahanadi | 19 | Daily | GHI |
| Peninsular India | Mahi | 6 | Daily | GHI |
| Peninsular India | Narmada | 18 | Daily | GHI |
| Peninsular India | Pennar | 6 | Daily | GHI |
| Peninsular India | Sabarmati | 2 | Daily | GHI |
| Peninsular India | Subernarekha | 4 | Daily | GHI |
| Peninsular India | Tapi | 4 | Daily | GHI |
| Peninsular India | WFR North | 4 | Daily | GHI |
| Peninsular India | WFR South | 24 | Daily | GHI |

minimal missing records. A total of 242 stations (and their watersheds) are used in this analysis, with 213 of these watersheds being from Peninsular India and 29 from Northern India (dots in Figure 3 (panel (a)). The number of stations within each basin and other pertinent information is summarized in Table 1.

## 115 2.2 Precipitation

Select $P$ datasets used in this study are outlined in Table 2 and are briefly described here. In addition to these datasets, the PBCOR dataset is used as a reference climatology in certain parts of this analysis. Additional additional information on the PBCOR dataset is in Appendix A, while additional information on $P$ datasets is in Appendix B.

The $P$ datasets used here were often identified in the recent literature to be reasonable representation of observed $P$, and 120 range in spatial resolution from about 4 km to 25 km, and temporal frequency of half hour to a month. Datasets included here are based on rain gauges (e.g., IMD), or reanalysis (e.g., ERA5-Land), or satellites, or a combination of sources (e.g., CHIRPS). The IMD gauge-based dataset is of primary interest in this study since it is the often used benchmark in a number

**Table 2.** Precipitation datasets used in this study and relevant information. Input used in the creation of each dataset is indicated by one or more of 'G' (gauge), 'O' (observation-based data product), 'R' (reanalysis) and 'S' (satellite). Ending year in the time span is left blank when a dataset extends to the present.

| Product (Version) | Alias | Input(s) | Native Resolution | Reference | Time Span |
|---|---|---|---|---|---|
| APHRODITE (v1101)[a] | APHRO | G,O | Daily, 0.25 deg (~25 km) | Yatagai et al. (2012) | 1951-2015 |
| CHIRPS (v2)[b] | CHIRPS | S,R,O | Daily, 0.05 deg (~5 km) | Funk et al. (2014) | 1981- |
| CPC CMORPH (v1)[c] | CMORPH | S,O | 0.5 Hourly, 8 km | Xie et al. (2017) | 1998- |
| ERA5-Land[d] | ERA5 | R | Hourly, 0.10 deg (~10 km) | Muñoz-Sabater et al. (2021) | 1950- |
| GSMaP (v6, Gauge_NRT)[e] | GSMAP | S,O | Hourly, 0.10 deg (~10 km) | Kubota et al. (2020) | 2000- |
| IMD | IMD | G | Daily, 0.25 deg (~25 km) | Pai et al. (2014) | 1950- |
| IMD/APHRODITE blend | IMD-APHRO | | Monthly, 0.25 deg (~25 km) | This study, Section 2.2 | 1951-2015 |
| IMDAA[f] | IMDAA | R | Hourly, 0.12 deg (~12 km) | Rani et al. (2021) | 1980- |
| IMERG (Final, v06B)[g] | IMERG | S,O | 0.5 Hourly, 0.10 deg (~10 km) | Huffman et al. (2020) | 2000- |
| MSWEP (v2, Past_nogauge)[h] | MSWEP | S,R,O | 3 Hourly, 0.10 deg (~10 km) | Beck et al. (2019) | 1980- |
| PERSIANN (CCS-CDR)[i] | PERSIANN | S,O | 3 Hourly, 0.04 deg (~4 km) | Sadeghi et al. (2021) | 1983- |
| SM2RAIN (ASCAT, v1.5)[j] | SM2RAIN | S,O | Daily, 0.10 deg (~10 km) | Brocca et al. (2019) | 2007- |
| TerraClimate | TERRA | R,O | Monthly, 0.042 deg (~4 km) | Abatzoglou et al. (2018) | 1958- |

[a] Asian Precipitation - Highly-Resolved Observational Data Integration Towards Evaluation of Water Resources; [b] Climate Hazards Group InfraRed Precipitation with Station data; [c] Climate Prediction Center Morphing Technique; [d] European Centre for Medium-Range Weather Forecasts (ECMWF), land component of the fifth generation of European ReAnalysis (ERA5); [e] Global Satellite Mapping of Precipitation; [f] Indian Monsoon Data Assimilation and Analysis reanalysis; [g] Integrated Multi-satellitE Retrievals for Global Precipitation Measurement; [h] Multi-Source Weighted-Ensemble Precipitation; [i] Precipitation Estimation from Remotely Sensed Information using Artificial Neural Networks (PERSIANN) - Cloud Classification System (CCS) - Climate Data Record (CDR); [j] Soil Moisture to Rain (SM2RAIN) Advanced Scatterometer (ASCAT).

of studies. Only those datasets whose spatial resolution is the same or finer than IMD's resolution of 0.25 deg (~25 km) were chosen here.

The reader should note that while the IMD dataset is limited to India's political boundaries, the rest of the $P$ datasets are not. However, certain river basins of India extend beyond India's boundaries and are part of this analysis. To enable an appropriate comparison between datasets, the IMD dataset is complemented, where needed, with the APHRODITE dataset (Yatagai et al., 2012). The APHRODITE dataset was chosen for several reasons: it is also based on rain gauge data, similar to IMD; its spatial and temporal resolution are the same as IMD's resolution (0.25 deg or ~25 km, and daily); and studies in the literature

have found that APHRODITE compares reasonably with IMD across many parts of India (e.g., Prakash et al., 2015b). While limitations with APHRODITE are discussed by such studies, it is assumed to be the best gauge-based alternative to IMD.

    For those regions where IMD's data is unavailable, grids from APHRODITE were identified, then the data from such grids was interpolated to align with the IMD grid. Finally, a blended product called IMD-APHRO which spanned the entire study domain was created. In the remainder of this paper, unless otherwise stated, IMD-APHRO refers to the blended product created

**Table 3.** Evapotranspiration datasets used in this study and relevant information. Ending year in the time span is left blank when a dataset extends to the present.

| Product (Version) | Alias | Native Resolution | Reference | Time Span |
|---|---|---|---|---|
| NTSG/PLSH[a] | NTSG | Monthly, 0.083 deg (~8 km) | Zhang et al. (2010) | 1982-2013 |
| GLEAM (v3.6a)[b] | GLEAM | Daily, 0.25 deg (~25 km) | Martens et al. (2017), Miralles et al. (2011) | 1980- |

[a] Numerical Terradynamic Simulation Group (NTSG) Process-based Land Surface Evapotranspiration/Heat (PLSH) Fluxes Algorithm; [b] Global Land Evaporation Amsterdam Model.

here, and IMD refers to the product confined to India's political boundaries. Also, in the remainder of this paper, each $P$ product is referred to by its 'Alias' in Table 2.

## 2.3 Evapotranspiration

A number of $ET$ datasets are currently available and the reader is referred to Zhang et al. (2016) and Karimi and Bastiaanssen (2015) for a review of such datasets. Two datasets were considered for this analysis based on their usage in studies across India (Table 3) - NTSG and GLEAM. Global Land Evaporation Amsterdam Model (GLEAM) provides estimates of the different components of $ET$, including transpiration, bare-soil evaporation, interception loss, open-water evaporation and sublimation (Martens et al., 2017; Miralles et al., 2011). A comparison of NTSG and GLEAM datasets (Appendix C) indicates that they are generally consistent with each other across several basins. However, estimates from GLEAM tend to be lower than those from NTSG. GLEAM was the primary dataset used in this study because of its longer time span and its availability to the present time.

## 2.4 Other Data

### 2.4.1 Elevation, Land Cover and Land Use

Figure 4 (panel (a)) shows the spatial variability in elevation across the study domain. The dominant features include the Himalayas in the Northern and Northeastern parts of the study domain, the mountains (or Ghats) along the Western and Eastern Coasts of India, the plains of the Ganga and Brahmaputra basins, and the Deccan Plateau in Peninsular India.

Land cover datasets from Indian agencies, such as those from NRSC (2007), are limited to India's political boundaries. Hence, the high resolution (100 m) global dataset based on PROBA-V satellite (Buchhorn et al., 2020) was used to identify the dominant land cover and land use types. Figure 4 (panel (b)) shows the spatial distribution of the dominant land cover and land use. For the purpose of this analysis, the land cover types of grass, shrub and tree/forest were pooled into one category, assuming that such land cover has minimal presence of humans and can be considered closest to natural conditions. From Figure 4 (panel (b)) it is evident that a vast majority of this pooled category is present in the mountainous regions of Northern India, Western coast or Central India. The rest of the study domain is predominantly covered by crops. It is also evident that

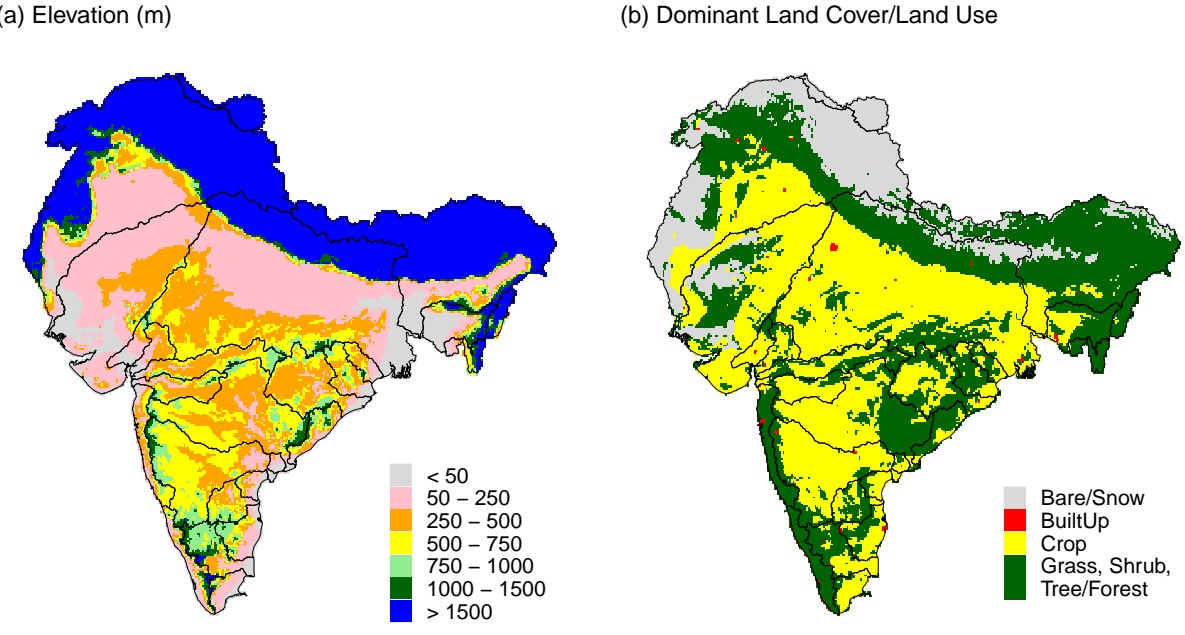

(a) Elevation (m)

(b) Dominant Land Cover/Land Use

Elevation legend:
< 50
50 – 250
250 – 500
500 – 750
750 – 1000
1000 – 1500
> 1500

Land cover legend:
Bare/Snow
BuiltUp
Crop
Grass, Shrub, Tree/Forest

**Figure 4.** (a) Elevation based on HydroSHEDS 500 m topographic data. For ease of display, 500 m data was aggregated to a 10 km resolution and the maximum values within each 10 km grid is displayed. (b) Major land cover and land use types based on PROBA-V 100 m data. For ease of display, 100 m data was aggregated to a 10 km resolution and the dominant value within each 10 km grid is displayed.

many of the higher elevation regions are covered by the pooled category of grass, shrub and tree/forest and the lower elevation regions are covered by crops.

### 2.4.2 Water Management

Water management considered in this study includes groundwater extraction, diversions (imports and exports) and reservoir storage and are summarized in Figure 5. A detailed description of this data is in Appendix D and only a short overview is presented here. Groundwater extraction and recharge estimates are available from the India's Central Ground Water Board (CGWB) for select years. The extent of annual groundwater extraction is quantified as a fraction of the annual $P$. Similarly, basin-scale imports and exports from CWC-19 (2019), were expressed as a fraction of annual $P$. Information on large dams and reservoirs in India was obtained from the National Registry of Large Dams (NRLD, 2019). For each of the 242 watersheds

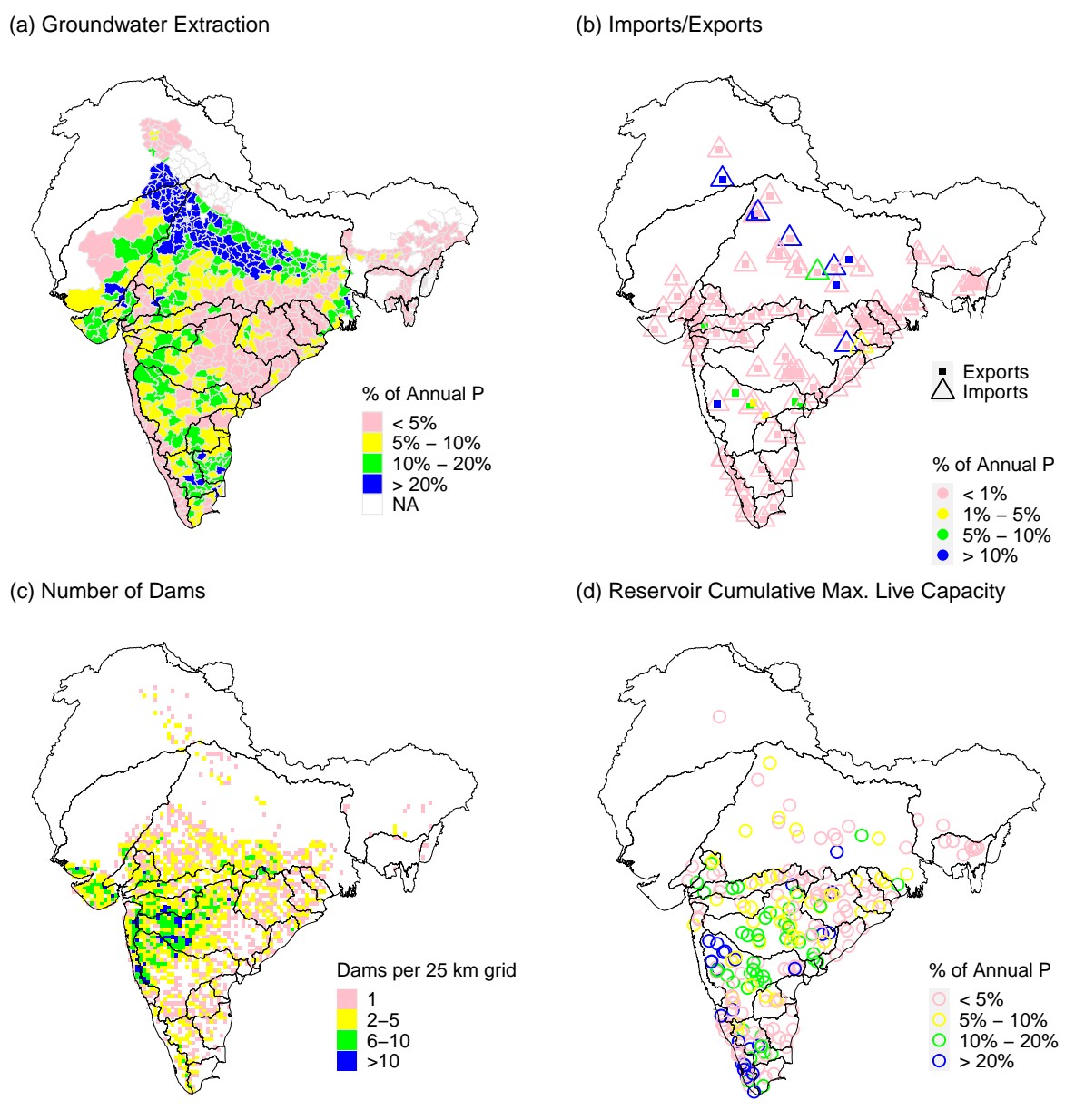

**Figure 5.** (a) District-wise annual groundwater extraction as a percent of annual precipitation; (b) basin-scale imports and exports as fraction of annual $P$; (c) density of dams and reservoirs, represented as number of dams per 0.25 deg (about 25 km) grid; (d) cumulative maximum live storage capacity of each watershed expressed as a fraction of annual $P$, for WY 2019.

used in this study, the cumulative live storage capacity from all dams present within the watershed was expressed as a fraction of annual $P$.

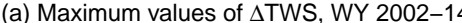

(a) Maximum values of ΔTWS, WY 2002–14

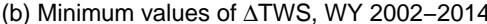

(b) Minimum values of ΔTWS, WY 2002–2014

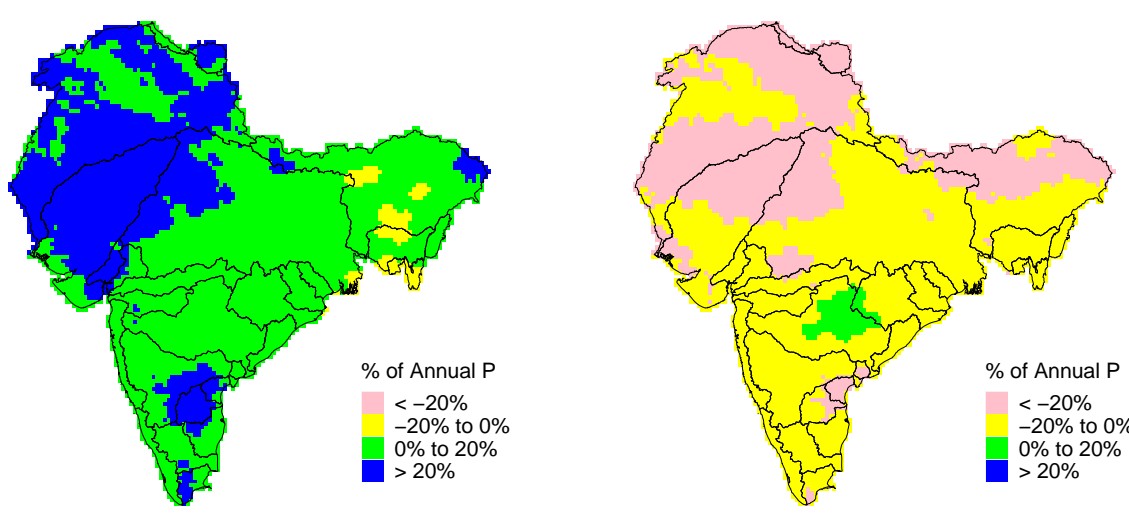

**Figure 6.** (a) Grid-wise maximum values of annual $\Delta TWS$ for WY 2003-2014. (b) same as (a) but for the minimum values.

Figure 5 (panel (a)) shows that groundwater extraction can be a substantial fraction of annual $P$ in certain parts of India,
but is minimal in the mountainous and wet regions of India - Western Coast of India, Northernmost and Northeastern India.
Similarly, water diversions are highest in the agricultural regions of the Ganga Basin and the interior parts of Peninsular India.
The highest density of dams is in the arid Western India, while the lowest density is in the plains and mountains of Northern
India. There are some watersheds in the Coastal Peninsular India where reservoir storage is a significant portion of the annual
$P$, but most of the other watersheds are minimally impacted by such storage.

### 2.4.3 Changes in Terrestrial Water Storage (TWS)

Changes in terrestrial water storage ($TWS$), inferred from the Gravity Recovery and Climate Experiment (GRACE) satellite
mission (Tapley et al., 2004), are valuable for many reasons, including identifying regions where large-scale water management
is causing substantial changes to the natural hydrologic cycle (e.g., Famiglietti, 2014; Rodell et al., 2009). $TWS$ includes water
stored below the ground, on the ground and above the ground. Thus, groundwater, soil moisture, surface water (rivers, lakes,
wetlands and reservoirs), snow and ice (including glaciers), canopy interception, and water within vegetation are all included
within $TWS$. GRACE-based $TWS$ anomalies from the Center for Space Research (CSR) (Save et al., 2016; Save, 2020) were
used to estimate change in annual $TWS$ (or $\Delta TWS$) as a fraction of the annual $P$ (see Appendix E).

Figure 6 shows the maximum $\Delta TWS$ (panel (a)) and minimum $\Delta TWS$ (panel (b)) over the period WY 2002-2014. The
magnitude of such changes for most of the study domain is within +20% or -20% of annual $P$. However, there are regions
such as Northwestern, Northern and Eastern India, where the magnitude of such changes is larger than 20% of annual P. The

reader should note that GRACE-based $\Delta TWS$ presented here has limitations and may not always adequately represent actual $\Delta TWS$. A further discussion on such limitations is in Section 5.1.1.

## 3    Methods

In this study $UoP$ is said to occur when observed annual precipitation ($P_{obs}$) is less than the actual annual precipitation ($P_{act}$)
when averaged over the entire watershed (Eq. 1). Since $P_{act}$ is not known, expected empirical relationships between $P_{obs}$ and other hydrological fluxes are examined to identify gross $UoP$. Watersheds affected by $UoP$ would be those where the balance between inputs ($P_{obs}$) and outputs (e.g., $R$ and $ET$) cannot be reconciled, despite reasonably accounting for changes in $TWS$ or disruptions to the natural balance caused by large-scale management. Watersheds are assumed to have negligible flow across topographic boundaries - i.e., inter-watershed groundwater flow is ignored. The particular $UoP$ scenarios analyzed here are
described in Section 3.1. The methodology used to compile the needed data for such an analysis is described in Section 3.2.

$$P_{obs} < P_{act} \tag{1}$$

### 3.1    Water Imbalance Scenarios

In order to take advantage of the datasets on $TWS$ anomalies and water management discussed in Section 2.4, the traditional annual water balance equation is formulated in two different ways in the following discussion.
Under natural circumstances, one could express the annual water balance of a watershed by assuming that the net change in terrestrial water storage ($TWS$) is the imbalance between total actual precipitation ($P_{act}$), the output fluxes of $R$ and $ET$ and groundwater flow across topographic boundaries, or inter-watershed ground water flow (IGF) (Eq. 2). $TWS$ is the sum of all the potential water reservoirs - groundwater, soil moisture, snow water equivalent, surface water, land ice, and water in the biomass (Humphrey et al., 2023). Watershed boundaries do not always coincide with underlying aquifer boundaries, and
IGF could play an important role in the watershed's water balance (e.g., Fan, 2019; Liu et al., 2020). However, in the absence of field data on groundwater flow pathways it is not possible to quantify the effect of IGF on the water balance. This study assumes IGF to be negligible. The implications of this assumption are discussed in Section 5.1.2.

$$\Delta_{TWS}^{natural} = P_{act} - R - ET + \Delta_{IGF} \tag{2}$$

If one were to account for the effects of management, $\Delta_{TWS}$ would represent changes due to both natural and human-
related causes such as groundwater extraction, reservoir storage and diversions. Under such circumstances, after ignoring IGF, one could reformulate Eq. 2 as Eq. 3. Net surface water diversions are represented by the terms $Exports$ (net loss of water) and $Imports$ (net gain of water). The terms $P_{act}$, $R$, $ET$, $Exports$ and $Imports$ are non-negative. $\Delta_{TWS}$ is positive if there is a net increase in $TWS$ and negative if there is a net decrease in $TWS$.

$$\Delta_{TWS} = P_{act} - R - ET + Imports - Exports \tag{3}$$

Rearranging Eq. 3 results in Eq. 4. The equality in Eq. 3 has been replaced with an approximation in Eq. 4 because the needed data, if available, is often not at the spatial or temporal resolution required to accurately balance the water budget.

$$P_{act} \approx R + ET + \Delta_{TWS} + Exports - Imports \tag{4}$$

     There is another way, although more approximate than Eq. 4, of formulating the annual water balance. Management of water is present in many parts of India, and includes groundwater extraction, reservoir storage and diversions (CWC-19, 2019). To
take advantage of such data on management (Section 2.4), the annual water balance is approximated as Eq. 5. Groundwater storage changes from both natural ($\Delta_{GW\ natural}$) and human-caused changes ($\Delta_{GW\ human}$) are included. Changes to reservoir storage ($\Delta_{Reservoir}$) and diversions ($Exports$ and $Imports$) are also explicitly included. In Eq. 5, both $\Delta_{GW}$ terms are positive if there is a net aquifer recharge, and negative if there is a net aquifer depletion. Thus, groundwater extraction presented in Figure 5 would be a negative quantity. $\Delta_{Reservoir}$ is positive if there is a net increase in reservoir storage and negative if
there is a net decrease in storage.

$$P_{act} \approx R + ET + \Delta_{GW\ natural} + \Delta_{GW\ human} + \Delta_{Reservoir} + Exports - Imports \tag{5}$$

     The reader should note that Eq. 4 and Eq. 5 are two separate, but useful, ways of analyzing the water budget. While $\Delta_{TWS}$ in Eq. 4 includes changes in all potential water reservoirs, Eq. 5 is an approximation and does not adequately capture the effect of snow processes, does not include water stored as soil moisture, and does not capture all the effects of management. The
reader should also note that often hydrologic analyses make the *a priori* assumption of net annual change in storage ($\Delta_{TWS}$ or $\Delta_{GW}$) being negligible. This study does not make such an assumption within Eq. 4 and Eq. 5.

     If $UoP$ is absent (i.e., $P_{obs} \approx P_{act}$), then based on Eq. 4 and Eq. 5 it is reasonable to expect $R$ to be only a portion of $P_{act}$, regardless of the extent of management. If the effects of management - the two rightmost terms of Eq. 4, and the four rightmost terms of Eq. 5, are relatively small compared to $P_{act}$, then it is also reasonable to expect $P_{act}$ to approximately equal
$R + ET + \Delta$, where $\Delta$ is either $\Delta_{TWS}$ or $\Delta_{GW\ natural}$. As discussed later in this Section, for most watersheds of the study domain, a reasonable upper bound on the magnitude of $\Delta_{TWS}$ (and $\Delta_{GW\ natural}$) is 20% of $P_{obs}$. The above expectations are illustrated by the 'Likely Scenarios' in Figure 7.

     If $UoP$ is present (i.e., $P_{obs} << P_{act}$), one could potentially realize the 'Spurious Scenarios' of $P_{obs} \leq R$ and $P_{obs} << R + ET$ (see Figure 7), when the extent of management is minimal. If, on the other hand, management is moderate to extensive,
it is difficult to generalize the relationship between the relative magnitudes of $P_{act}$, $R$ and $ET$, since $R$ and $ET$ are no longer constrained by the natural water balance. The 'Spurious Scenarios' in this situation include only the case of $P_{obs} \leq R$. The reader should note that 'minimal' is used in a relative sense when the overall effect of annual management at a watershed-scale relative to annual $P_{obs}$ is minimal. It should not be interpreted as the effect of local management on specific storm events.

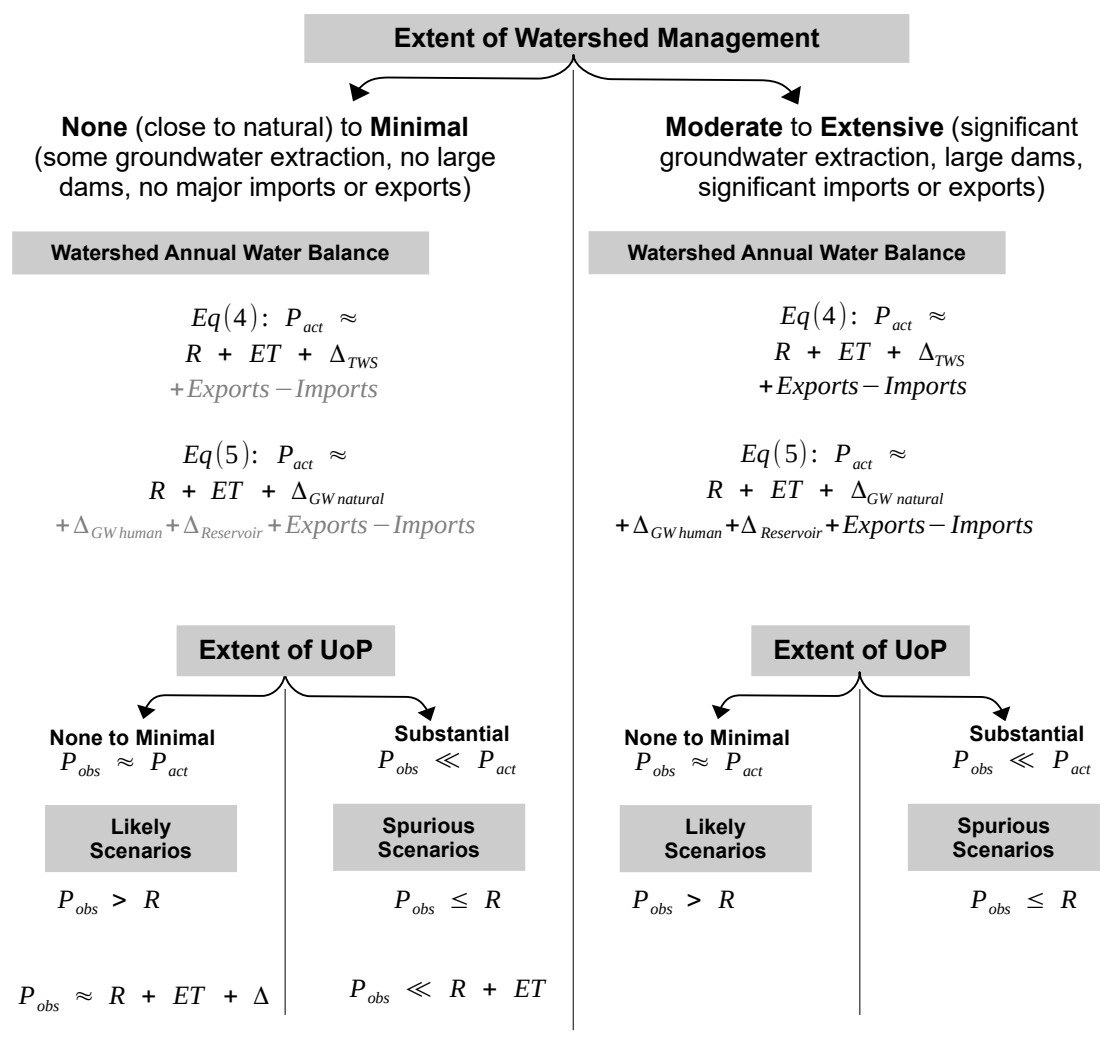

**Figure 7.** Schematic illustrating potential spurious scenarios derived from Eq. 4 and Eq. 5, when $P_{act}$ is underestimated. The terms in grey text within the annual water balance equations are relatively small compared to $P_{act}$.

The two specific scenarios investigated in this study are formulated via Eq. 6 and Eq. 7, and are based on the 'Spurious Scenarios' in Figure 7. In Scenario I, annual $P_{obs}$ is less than or equal to $R$, i.e., annual runoff coefficient is at least 1.0. This scenario could be realized regardless of the extent of management outlined in Figure 7. Such a scenario was also used by other studies (e.g., Beck et al., 2020) to identify $UoP$. However, instances where the annual runoff coefficient is less than 1.0, but still spuriously high (e.g., 0.95), are excluded by Scenario I. Scenario II attempts to include such instances. Moreover, Scenario II is also intended to capture instances where the sum of $R$ and $ET$ greatly exceeds $P_{obs}$. If $UoP$ is present, relatively high values of $R$ when combined with reasonable estimates of $ET$ result in the sum of $R$ and $ET$ greatly exceeding $P_{obs}$.

The formulation of Scenario I exactly follows the first of the 'Spurious Scenarios' in Figure 7. The second of the 'Spurious Scenarios' in Figure 7, $P_{obs} << R + ET$, is not an exact mathematical relationship. The formulation of Scenario II makes such a relationship exact by the use of heuristics. The rationale behind such heuristics is presented in the following discussion.

$$Scenario\ I:\ P_{obs} \leq R \tag{6}$$

$$Scenario\ II:\ (0.70 \times P_{obs}) \leq R < P_{obs}\ \&\ (1.20 \times P_{obs}) \leq R + ET \tag{7}$$

The typical wet season runoff coefficient for the whole of India was estimated to be about 0.38 by Gupta et al. (2016). The basin-scale average annual runoff coefficient was estimated by Xiong et al. (2022) to range from 0.10 to 0.40 for several large river basins of India, with higher coefficients for the Indus and Brahmaputra basins. Considering the magnitude of such estimated runoff coefficients, a coefficient of 0.70 was assumed to be a reasonable lower bound for identifying spuriously high annual runoff coefficients.

As shown in Figure 6, for regions having hilly terrain or covered by forests (Figure 4), the magnitude of $\Delta_{TWS}$ is typically within 20% of annual $P_{obs}$. Watershed management is represented by the four rightmost terms of Eq. 5: $\Delta_{GW\ net\ recharge}$, $\Delta_{Reservoir}$, $Exports$ and $Imports$. In the regions having hilly terrain or covered by forests (Figure 4), where management can be assumed to be minimal or non-existent, the magnitude of the individual effect of each type of management is typically less than 5% of $P_{obs}$ (Figure 5, panels (a), (b) and (d)). A reasonable upper bound on the cumulative effect of the four rightmost terms of Eq. 5 is also 20% of annual $P_{obs}$. Thus, when management can be considered minimal or non-existent, it is reasonable to expect $R + ET$ to have maximum value of $1.20 \times P_{obs}$. This is the justification for the heuristic of 1.20 in Eq. 7. As mentioned earlier, 'minimal' management is used in the context of overall effect of annual management at a watershed-scale relative to annual $P_{obs}$. For instance, a 20% management effect of annual $P_{obs}$, in a watershed with 0.4 as the runoff coefficient, translates to 50% (20% / 0.4 = 50%) effect on annual $R$. Thus, 'minimal' management could still have a substantial effect on the annual $R$.

This study identifies $UoP$ by first identifying individual years within watersheds where Eq. 6 and Eq. 7 are realized. Then, it proceeds to investigate the extent of management and extent of $\Delta_{TWS}$ within such imbalanced watersheds. If management and $\Delta_{TWS}$ are deemed minimal relative to annual $P$, then it is concluded that the likely cause of such spurious imbalance is

$UoP$. The two scenarios are formulated to be non-overlapping. The formulation of these two scenarios and the heuristics used within them are subjective and can affect the results of the analysis. Moreover, the formulation of the scenarios relies more on $R$ and less on $ET$. This is because observed $R$ is assumed to be more reliable than satellite-based $ET$. A further discussion on this is in Section 5.1.2.

## 3.2   Time Series Compilation

In order to investigate the above mentioned scenarios, annual time series of all the terms of Eq. 6 and Eq. 7 need to be compiled. All needed variables are expressed in the same units of volume. Observed daily streamflow ($R$), available in units of $\mathrm{m}^3/\mathrm{s}$ was aggregated to cumulative monthly and annual volumes ($\mathrm{MCM/month}$ and $\mathrm{MCM/year}$, respectively). Gridded data on $P_{obs}$ and $ET$, available in units of depth per unit area per month (e.g., $\mathrm{mm/month}$), were also aggregated to watershed-scale monthly and annual volumes. The process of aggregating grid-based products to a watershed involves identifying the

spatial overlap between the grids and the watershed. Such relationships were identified using a Geographic Information System (GIS) analysis. Grid-specific fractional areas were used in the process of aggregation. A schematic illustrating the process of aggregation is in Figure 8. The needed time series were compiled for each of the 242 watersheds analyzed in this study. $P$ datasets are often available up to the current year, but the latest year for which observed $R$ is available is WY 2017, and $ET$ is available since WY 1980. The time span of the data compiled in this study is WY 1980 to 2017 (38 WYs), whenever data is

available.

## 4   Results

The results presented here follow the specific objectives outlined in Section 1.1. Observed $UoP$ within the IMD-APHRO dataset is discussed in Section 4.1, including an example illustrating the spurious water imbalance potentially caused by $UoP$, and the spatial extent of the imbalanced watersheds. The hydroclimatological characteristics of such imbalanced watersheds,

including the extent of management, are discussed in Section 4.2. Extent of $UoP$ within all $P$ datasets is compared in Section 4.3. Using select datasets which experienced $UoP$ to a lesser extent than IMD-APHRO, grid-wise potential correction factors (CFs) associated with IMD were estimated in Section 4.4. Basin-scale potential CFs are also discussed in Section 4.4.

## 4.1   Imbalanced Watersheds using IMD-APHRO

An example showing the annual water imbalance scenario of $P \leq R$ is in Figure 9, for the Bantwal station on Nethravathi river

in the WFR South Basin. The annual time series is in panel (a) while the monthly time series for select years is in panel (b). There are several WYs where the total annual volume of $P$ is less than total observed $R$, such as WYs 2011-13 in the recent past. The total live capacity of all upstream reservoirs is 0 since there are no dams present in this watershed. The monthly time series is also shown in for select years (WYs 2011-15, panel (b)). The strong seasonal pattern imposed by the Summer monsoon is evident, with the months of June-September having the highest values of $P$ and $R$. There are several months within

each year where observed $R$ is greater than $P$. It is useful to note that the above spurious relationship of annual $R$ exceeding

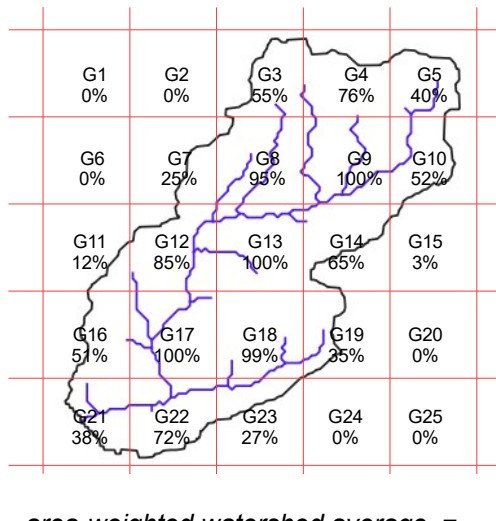

$$\text{area-weighted watershed average} = \frac{\displaystyle\sum_{g=1}^{g=25} Variable \times Area_g \times Fraction_g}{\displaystyle\sum_{g=1}^{g=25} Area_g \times Fraction_g}$$

**Figure 8.** Schematic illustrating the area-weighted averaging procedure used to obtain watershed-aggregated variables from gridded products. The watershed is shown in black, the river network in blue, and the grid mesh in red. Grids are numbered G1 through G25.

annual $P$, for the Bantwal watershed, was also tabulated by CWC-19 (2019) (see their Appendix R, Table R-2), using the same $P$ and $R$ data sources as those used here.

Watersheds where either Scenario I or Scenario II was realized were identified by analyzing annual $P$, $R$ and $ET$ for all of the 242 watersheds included in this study. Figure 10 shows the catchment areas corresponding to these imbalanced watersheds (grey areas), and the gauging stations at the outlet of such watersheds (blue dots). These watersheds are located along the Western Coast of India, in the forested and hilly regions of Central India, and within the Himalayan mountains and their foothills. The locations of these imbalanced watersheds coincides with the regions receiving the highest annual $P$ (see Figure 1). Most river basins have at least one imbalanced watershed. Some of these watersheds have catchment areas that are outside of India's political boundaries. Such watersheds with at least 1% of the total catchment area outside of India are shown in pink in Figure 10. Due to the limited availability of observed $R$ data in Northern India, only a small number of imbalanced watersheds could be identified. In contrast, Peninsular India has many more imbalanced watersheds.

The watersheds identified above are based on a specific set of heuristics (0.70 and 1.20) used in Eq. 7. In order to understand the impacts of changing such heuristics, three other sets of heuristics were tried. Instead of 0.70, values of 0.60, 0.80 and 0.90 were used, and instead of 1.20, values of 1.10, 1.30 and 1.40 were used in Eq. 7. Figure S42 (Supplement) shows the resulting

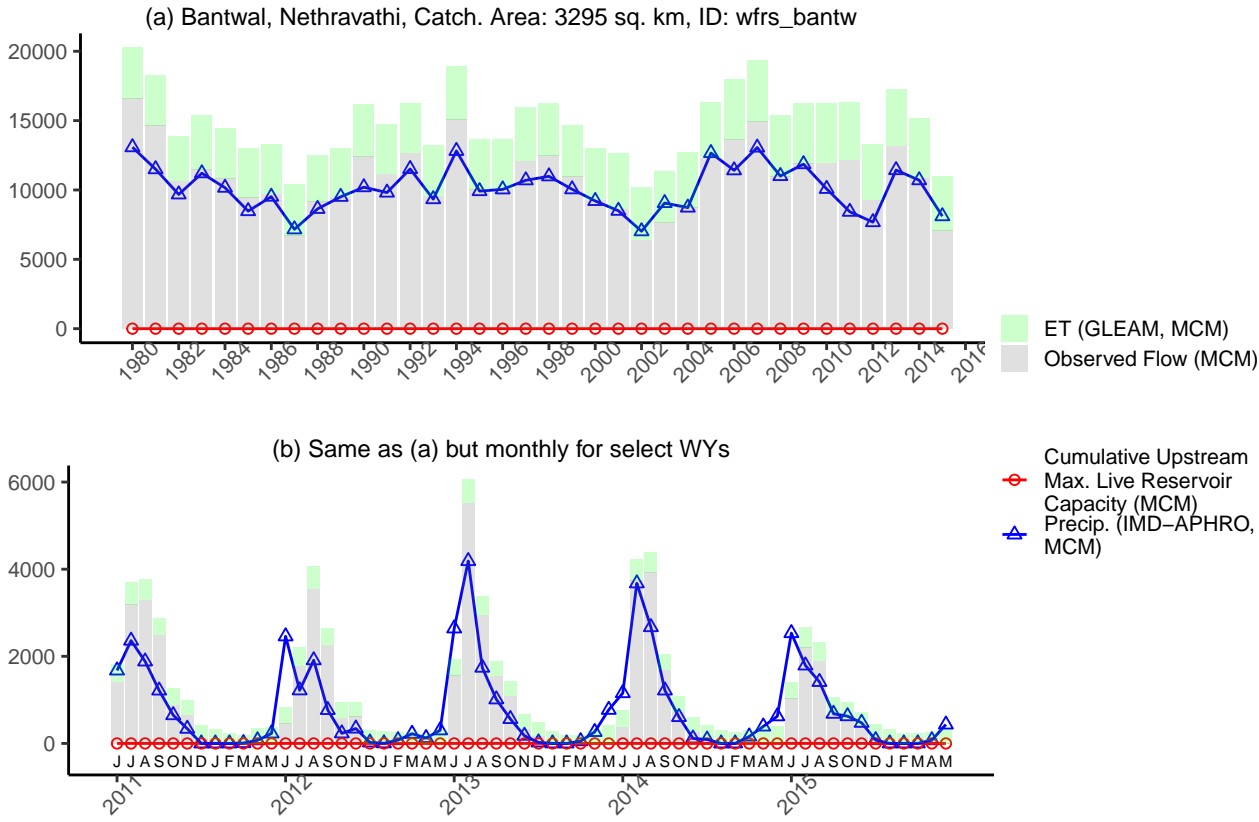

**Figure 9.** (a) Example showing the imbalance scenario of $P \leq R$ (Scenario I) for Bantwal station on Nethravathi river in the WFR South Basin. Annual $R$ (grey bars) exceeds annual $P$ (blue line) in certain years. $ET$ (green bars) and cumulative reservoir storage capacity (red line) are also shown for reference. (b) Monthly volumes instead of annual, for select WYs. Months are indicated by the first letter of their names, and follow the June-May WY convention.

imbalanced watersheds with each set of heuristics. By lowering these heuristics, one would expect more watersheds to be categorized as imbalanced, while raising them would result in fewer watersheds. As expected, lower values of the heuristics (e.g, 0.60 and 1.10) result in a larger number of watersheds, and higher values (e.g., 0.90 and 1.40) result in a lower number of watersheds, compared to the watersheds shown in Figure 10. However, the general location of these watersheds remains the same - the Western Coast of India, the forested and hilly regions of Central India, and the Himalayan mountains and their
foothills.

## 4.2   Characteristics of imbalanced Watersheds

The dominant physical characteristics associated with these imbalanced watersheds are summarized in Figure 11. The size of these watersheds can range from more than a 100,000 $\text{km}^2$ in the northern portion of the study domain to less than a 1,000

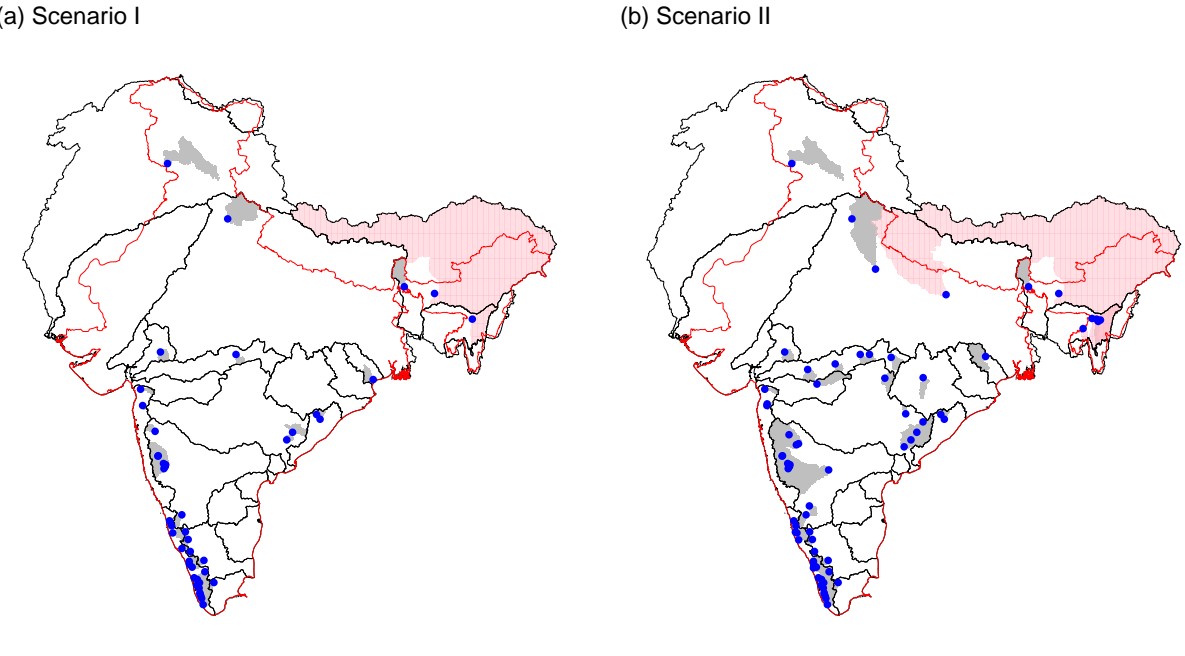

**Figure 10.** (a) Imbalanced watersheds from Scenario I (grey shaded regions) and the gauging stations (blue dots) at the outlets of such watersheds. Watersheds shaded in pink have at least 1% of their catchment area outside of India. (b) Same as (a) except for Scenario II.

$km^2$ in Peninsular India (panel (a)). The maximum elevation within such watersheds is about 2,000 m (panel (b)), much higher than the average elevation of India of about 600 m (estimated in this study). The statistics on fractional land cover and land use indicates that most of these watersheds are predominantly covered by natural land cover types (grass, shrub, tree/forest, or bare/snow) followed by crop (panel (c)).

Average annual $P$ for these imbalanced watersheds is typically around 2,000 mm/year (panel (d)), about twice the average annual $P$ for the whole of India (about 1,100 mm, see Table S1 in the Supplement). Thus, such watersheds are typically wetter than the rest of India. Moreover, what is presented here is observed $P$, potentially affected by $UoP$, and that the actual $P$ could be much higher. The maximum annual runoff coefficient for such watersheds typically exceeds 1.0 (median value of 1.15, maximum value of 3.33, panel (e)). The extent of reservoir storage is quantified as the cumulative sum of the maximum live storage capacity of all reservoirs present in the watershed, expressed as a percentage of average annual $P$ (panel (f)). While most watersheds have relatively minimal storage, some of them could have more than 50% of the annual $P$ captured in the reservoirs. However, the $P$ data used here is observed $P$, affected by $UoP$, and not actual $P$. Therefore, the actual effect

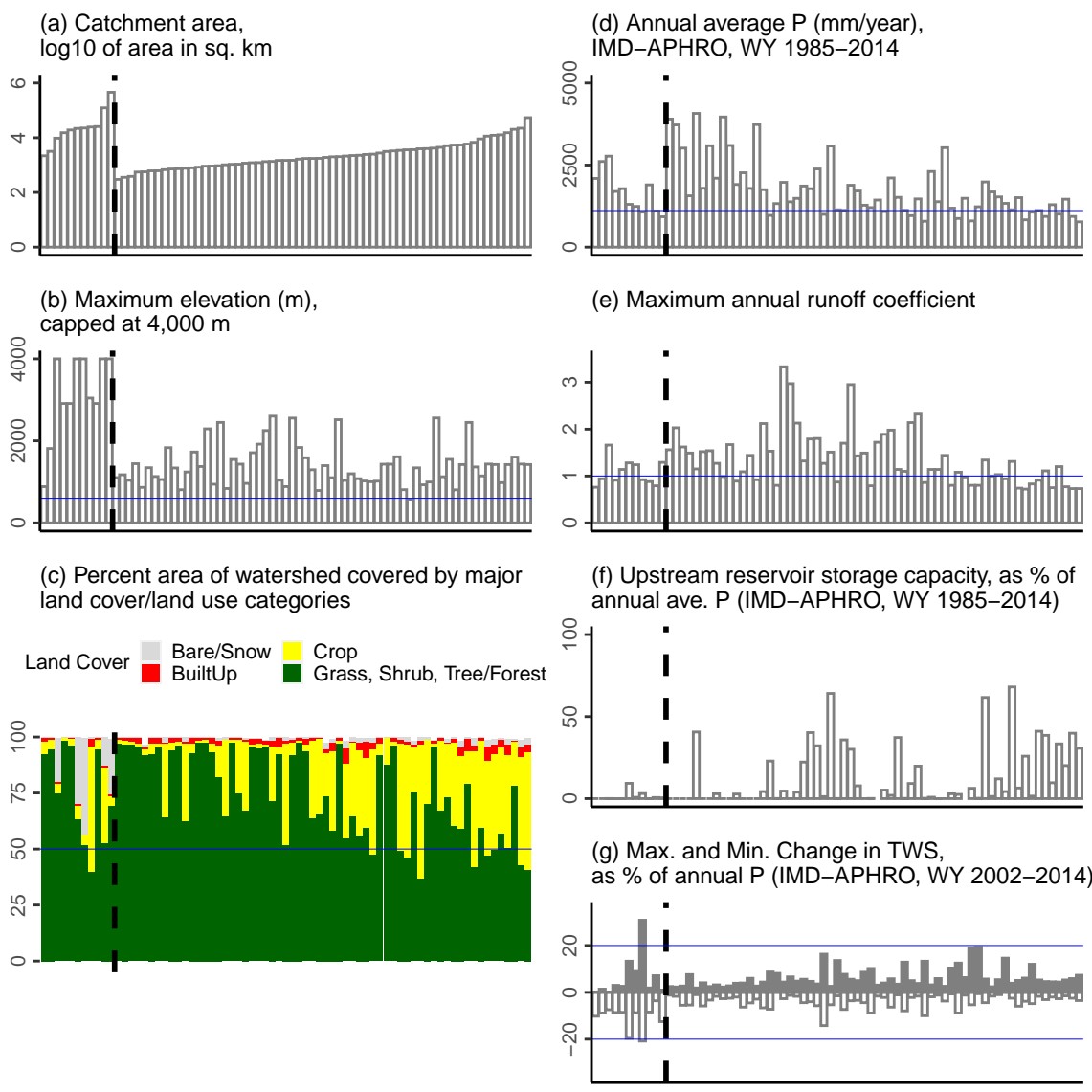

**Figure 11.** Characteristics of imbalanced watersheds identified using IMD-APHRO. The watersheds are sorted in ascending order of catchment area (left to right) in all of the panels. The vertical thick broken line separates the watersheds of Northern India from those of Peninsular India. (a) catchment area; (b) maximum elevation; blue line shows the average elevation for the whole of India (600 m); (c) land cover and land use fraction; blue line shows the 50% fraction for reference; (d) average annual $P$; blue line shows the average annual $P$ for the whole of India (1,100 mm); (e) maximum runoff coefficient; blue line indicates a value of 1.0; (f) cumulative maximum live storage capacity expressed as a fraction of annual $P$; and (g) maximum (shaded bars) and minimum (unshaded bars) $\Delta TWS$ as a fraction of annual $P$.

of reservoirs is expected to be smaller than what is represented here. Finally, minimum and maximum values of watershed-averaged values of $\Delta TWS$, expressed as fraction of annual $P$ (panel (g)), indicate that the magnitude of $\Delta TWS$ is less than 20% for most of these watersheds.

Based on these physical characteristics, the imbalanced watersheds identified using the IMD-APHRO dataset are typically forested (or minimally impacted by agriculture), present in relatively wet regions and in relatively high elevations, often have annual runoff coefficients exceeding 1.0, and, in general, minimally impacted by reservoir storage. Moreover, based on a visual comparison of the extent of large-scale management shown in Figure 5 and the locations of imbalanced watersheds in Figure 10, the imbalanced watersheds can be considered to be minimally affected by groundwater extraction and diversions. Furthermore, based on a visual comparison of annual $\Delta TWS$ in Figure 6 and the locations of imbalanced watersheds in Figure 10, the imbalanced watersheds are typically present in regions not affected by relatively large annual changes in $TWS$.

### 4.3 $UoP$ within IMD-APHRO versus Other Datasets

Similar to the earlier analysis of identifying watersheds potentially affected by $UoP$ using the IMD-APHRO dataset, potential $UoP$ within other $P$ datasets is analyzed in this Section. For each $P$ dataset, Table 4 shows the number of station-years across all imbalanced watersheds corresponding to that dataset. The number of station-years by scenario are tabulated separately for the watersheds of Northern India and Peninsular India. Since the different $P$ datasets have differing time spans, the total number of WYs varies by $P$ dataset. ERA5, IMD-APHRO and TERRA have the longest time span (782 station-years in Northern India and 6,153 station-years in Peninsular India) while SM2RAIN has the shortest time span (195 station-years in Northern India and 1,784 station-years in Peninsular India).

The total number of imbalanced years for which either $UoP$ scenario is realized is expressed as the percentage of the total analyzed station-years. Such a percentage acts as proxy for the extent of $UoP$, and can vary from about 2% to 29% in Northern India, and from 5% to 19% in Peninsular India, depending on the $P$ dataset. The APHRO dataset is consistent with IMD-APHRO in Peninsular India but not in Northern India. Across the entire study domain, the satellite-based GSMAP, PERSIANN and CMORPH datasets typically have the highest percentage of imbalanced station-years, while the reanalysis-based datasets of ERA5 and IMDAA have the lowest percentages. While ERA5 and IMDAA are consistent across both Northern and Peninsular India, the MSWEP and TERRA datasets have the lowest percentages in Peninsular India but do not have such low percentages in Northern India. The reanalysis-based datasets of ERA5 and IMDAA outperform IMD-APHRO as well as the high-resolution satellite products such as CMORPH and PERSIANN. The GSMAP dataset has the highest percentage of imbalanced watersheds in both Northern and Peninsular India.

The statistics presented in Table 4 are based on a specific set of heuristics (0.70 and 1.20) used within Eq. 7. In order to understand the impacts of changing such heuristics, three other sets of heuristics were tried. Instead of 0.70, values of 0.60, 0.80 and 0.90 were used, and instead of 1.20, values of 1.10, 1.30 and 1.40, were used in Eq. 7. Tables S3-S5 (Supplement) show the new set of statistics (similar to Table 4) for each set of heuristics. It is evident from these tables that the performance of the datasets remains similar to Table 4. ERA5 and IMDAA outperform IMD-APHRO consistently across both Northern and

**Table 4.** Total number of station-years analyzed for each $P$ dataset, and the imbalanced station-years by scenario, for all the watersheds of Northern India (left) and Peninsular India (right).

| | North India | | | | Peninsular India | | | |
|---|---|---|---|---|---|---|---|---|
| Product | Total | Scenario I | Scenario II | % Imbalanced | Total | Scenario I | Scenario II | % Imbalanced |
| APHRO | 782 | 92 | 73 | 21.1% | 5,788 | 349 | 385 | 12.7% |
| CHIRPS | 782 | 76 | 45 | 15.5% | 6,036 | 306 | 299 | 10.0% |
| CMORPH | 436 | 61 | 42 | 23.6% | 3,493 | 315 | 186 | 14.3% |
| ERA5 | 782 | 7 | 40 | 6.0% | 6,153 | 245 | 311 | 9.0% |
| GSMAP | 382 | 76 | 35 | 29.1% | 3,132 | 441 | 142 | 18.6% |
| IMD-APHRO | 782 | 32 | 59 | 11.6% | 6,153 | 414 | 372 | 12.8% |
| IMDAA | 782 | 8 | 9 | 2.2% | 6,153 | 179 | 239 | 6.8% |
| IMERG | 382 | 19 | 22 | 10.7% | 3,132 | 169 | 170 | 10.8% |
| MSWEP | 782 | 60 | 55 | 14.7% | 6,153 | 122 | 211 | 5.4% |
| PERSIANN | 782 | 89 | 60 | 19.1% | 5,796 | 767 | 260 | 17.7% |
| SM2RAIN | 195 | 9 | 13 | 11.3% | 1,784 | 116 | 79 | 10.9% |
| TERRA | 782 | 75 | 58 | 17.0% | 6,153 | 153 | 231 | 6.2% |

Peninsular India, while MSWEP and TERRA datasets have the lowest percentages in Peninsular India but do not have such low percentages in Northern India.

The metrics presented in Table 4 are associated with watersheds where adequate hydrometeorological data is available. Since such watersheds are limited to only certain portions of India, these metrics do not accurately reflect the spatial distribution of $UoP$ present within each $P$ dataset. In order to assess the spatial distribution of $UoP$, potential correction factors (CFs) are estimated for select datasets in Section 4.4. The ERA5, IMDAA, MSWEP and TERRA datasets are chosen for further analysis because of their potential ability to be less affected by $UoP$ than IMD-APHRO.

### 4.4 Potential Correction Factors (CFs) for Specific Datasets

Correction factors (CFs) represent the ratio of actual and observed $P$. Since it is not possible to estimate them without actual $P$, they were estimated assuming that select datasets from the above analysis are reasonable proxies for the actual $P$. Such estimated CFs are referred to as potential CFs to distinguish them from true CFs. As mentioned in Section 4.3, the ERA5, IMDAA, MSWEP and TERRA datasets suffer less from $UoP$ than IMD-APHRO. Using these datasets potential CFs were estimated using Eq. 8. For each dataset, data was first aggregated to IMD's resolution of 0.25 deg ($\sim$25 km). Then, for the 30-year common data period of WY 1985-2014, grid-wise average annual $P$ was estimated. The ratio of grid-wise 30-year average annual $P$ between each dataset and IMD is presented in Figure 12. The spatial domain is limited to the political boundaries of India where IMD data is available.

$$390 \quad CF^{dataset} = \frac{\overline{P}_{1985-2014}^{dataset}}{\overline{P}_{1985-2014}^{IMD}} \quad (8)$$

The spatial maps of potential CFs shown in Figure 12 can be compared to those presented in Figure 1 and those presented in Figure S1 of the Supplement. High CFs are present in the mountainous Western Coast of India for all the four datasets and in the mountainous parts of Northern India only for ERA5 and IMDAA. This is consistent with the percentage of imbalanced station-years associated with each of these datasets (see Table 4). Another feature that is evident from Figure 12 is that the

395 highest CFs occur in the wettest parts of India (see Figure 1, panel (a)). If these potential CFs are reasonably accurate, then one could conclude that $UoP$ is a substantial problem in the wettest parts of India. A CF of at least 1.5 (yellow shaded categories or higher in Figure 12) indicates that the actual $P$ is at least 50% higher than the observed $P$. There are wide swaths of mountainous and hilly regions of India with such CFs. In order to quantify which river basins of India are most affected by $UoP$, basin-aggregated potential CFs are analyzed.

Table 5 shows the basin-aggregated potential CFs for the above four $P$ datasets. An additional table for all of the $P$ datasets analyzed in this study is shown in Table S1 of the Supplement. The potential CFs shown here were estimated as the ratio of annual $P$ for each dataset and IMD. The average and maximum values for the 30-year period of WY 1985-2014 are shown in Table 5. Since IMD is the main $P$ dataset of interest and it is limited to the political boundaries of India, only that portion of each river basin falling within India's boundaries is included when estimating these potential CFs.

Across the whole of India, ERA5, IMDAA and MSWEP are on average, 9%, 26% and 3% higher than IMD, respectively, while TERRA is 2% lower than IMD. However, the maximum values indicate that, ERA5, IMDAA, MSWEP and TERRA can be up to 19%, 37%, 10% and 6% higher than IMD, respectively. There is substantial variability across basins and datasets. For instance, for the Brahmaputra Basin, ERA5 and IMDAA are 56% and 90% higher than IMD; however, MSWEP and TERRA are 5% and 9% lower than IMD. Similarly, for the Ganga Basin, on average, ERA5, IMDAA and MSWEP are 9%, 36% and

8% higher than IMD; however, TERRA is 1% lower than IMD. Similarly, for the Indus Basin, on average, ERA5 and IMDAA are 6% and 26% higher than IMD; however, MSWEP and TERRA are 33% and 43% lower than IMD. This pattern of ERA5 and IMDAA being higher than IMD, while MSWEP and TERRA being lower than IMD in the basins of Northern India is consistent with potential CFs shown in Figure 12. ERA5 and IMDAA have CFs exceeding 1.0 in many regions of Northern India, while MSWEP and TERRA do not have such high CFs to the same extent.

Table 5 also shows that for most basins of Peninsular India, potential CFs from the four selected $P$ datasets are almost always greater than 1.0. This implies that $P$ is underestimated across most of Peninsular India, regardless of which of the four datasets is used as a proxy for actual $P$. The Godavari and the Krishna basins are the two largest basins of Peninsular India. In the Godavari Basin, on average, the four datasets are 4% to 13% higher than IMD. In the Krishna Basin, on average, the four datasets are 13% to 19% higher than IMD. The wettest basins of Peninsular India are the WFR North and WFR South basins.

In these two basins, MSWEP and TERRA are higher than IMD, while ERA5 and IMDAA tend to be similar to or lower than IMD. This is consistent with the percentage of imbalanced station-years associated with each of these datasets in Peninsular India (see Table 4).

**Table 5.** Basin-aggregated potential CFs for select datasets based on the period WY 1985-2014. Average and maximum values for each dataset are shown for each river basin. The upper half of the table shows the basins of Northern India while the lower half shows the basins of Peninsular India.

| Basin | ERA5 | IMDAA | MSWEP | TERRA |
|---|---|---|---|---|
| All India | 1.09 / 1.19 | 1.26 / 1.37 | 1.03 / 1.10 | 0.98 / 1.06 |
| Barak | 0.87 / 1.24 | 1.20 / 1.61 | 0.98 / 1.31 | 0.98 / 1.80 |
| Brahmaputra | 1.56 / 1.97 | 1.90 / 2.51 | 0.95 / 1.19 | 0.91 / 1.18 |
| Ganga | 1.09 / 1.32 | 1.36 / 1.54 | 1.08 / 1.26 | 0.99 / 1.17 |
| Indus | 1.06 / 1.40 | 1.26 / 1.66 | 0.67 / 0.88 | 0.57 / 0.80 |
| Minor | 1.18 / 1.63 | 1.36 / 1.80 | 1.38 / 1.91 | 1.38 / 2.22 |
| North Ladakh | 0.70 / 1.83 | 0.45 / 1.33 | 0.38 / 1.03 | 0.06 / 0.19 |
| WFR Kutch | 0.93 / 1.04 | 0.98 / 1.54 | 0.97 / 1.13 | 0.94 / 1.12 |
| Brahmani-Baitarani | 1.00 / 1.22 | 1.13 / 1.42 | 1.04 / 1.23 | 1.00 / 1.20 |
| Cauvery | 1.21 / 1.95 | 1.28 / 1.98 | 1.17 / 2.00 | 1.13 / 2.13 |
| EFR North | 1.06 / 1.32 | 1.11 / 1.40 | 1.10 / 1.31 | 1.08 / 1.22 |
| EFR South | 1.05 / 1.88 | 1.27 / 2.11 | 1.11 / 2.04 | 1.10 / 2.30 |
| Godavari | 1.05 / 1.31 | 1.13 / 1.41 | 1.06 / 1.22 | 1.04 / 1.23 |
| Krishna | 1.13 / 1.27 | 1.16 / 1.34 | 1.19 / 1.35 | 1.16 / 1.52 |
| Mahanadi | 1.08 / 1.23 | 1.17 / 1.37 | 1.09 / 1.21 | 1.04 / 1.21 |
| Mahi | 1.00 / 1.29 | 0.96 / 1.35 | 1.02 / 1.28 | 1.03 / 1.63 |
| Narmada | 1.07 / 1.47 | 1.08 / 1.51 | 1.09 / 1.35 | 1.05 / 1.29 |
| Pennar | 0.96 / 1.24 | 1.18 / 1.59 | 0.95 / 1.20 | 0.94 / 1.33 |
| Sabarmati | 0.90 / 1.21 | 0.83 / 1.09 | 0.97 / 1.28 | 0.98 / 1.28 |
| Subernarekha | 0.97 / 1.11 | 1.12 / 1.46 | 1.03 / 1.20 | 0.99 / 1.30 |
| Tapi | 1.15 / 1.42 | 0.98 / 1.22 | 1.09 / 1.46 | 1.03 / 1.30 |
| WFR North | 0.73 / 1.07 | 0.77 / 1.15 | 1.10 / 1.54 | 1.13 / 1.55 |
| WFR South | 0.99 / 1.17 | 1.04 / 1.21 | 1.25 / 1.57 | 1.21 / 1.70 |

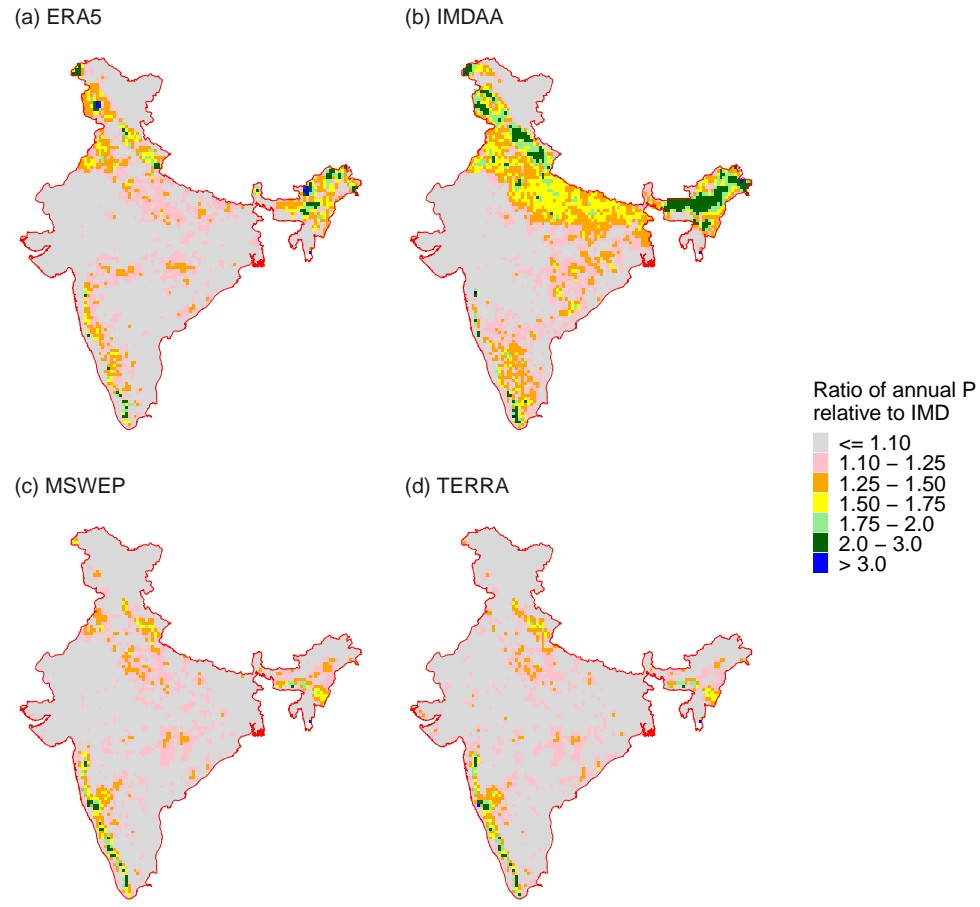

**Figure 12.** Grid-wise potential CFs for select datasets based on the period WY 1985-2014.

## 4.5 Summary of Results

The first objective of this study was to identify watersheds affected by $UoP$ using the IMD-APHRO dataset. Across many
watersheds of Northern and Peninsular India, Scenario I and Scenario II were realized. Such imbalanced watersheds span most
of the river basins of India and are typically located along the mountains of the Western Coast of India, in the forested and hilly
regions of Central India, and in the Himalayan mountains and their foothills. These regions also happen to be the relatively
wet regions receiving the most $P$. Moreover, the years in which Scenario I or II were realized span the entire historical record,
including the decade of 2010s, the most recent decade with available data.

The second objective was to investigate if the imbalance in these imbalanced watersheds is due to large-scale management
or due to relatively large changes in $\Delta TWS$ in such watersheds. It was found that the imbalanced watersheds typically have
forests as the dominant land cover ('Grass, Shrub, Tree/Forest' land cover category), and are present in relatively wet regions

and in relatively high elevations. Based on maps compiled in this study on large-scale management, the imbalanced watersheds are minimally affected by groundwater extraction and diversions (imports or exports). While some of these watersheds have the capacity to store a substantial portion of annual $P$ as reservoir storage, other watersheds have minimal reservoir storage. Using GRACE-based annual $\Delta TWS$ it is concluded that annual changes in $TWS$ in these imbalanced watersheds are relatively small compared to annual $P$. Thus, it is unlikely that management or annual $\Delta TWS$ are the causes of the imbalance in the imbalanced watersheds.

The third objective was to compare the extent of $UoP$ within IMD-APHRO against that in other $P$ products. The reanalysis-based datasets of ERA5 and IMDAA consistently outperform IMD-APHRO across both Northern and Peninsular India. The MSWEP and TERRA datasets outperform IMD-APHRO in Peninsular India but not in Northern India. However, none of the datasets analyzed here are completely free of $UoP$. If these four datasets are to be considered as alternatives to IMD when addressing $UoP$, based on the 30-year period of WY 1985-2014, averaged across the whole of India, ERA5, IMDAA and MSWEP are 9%, 26% and 3% higher than IMD, respectively, while TERRA is 2% lower than IMD. However, there are years within this 30-year period when ERA5, IMDAA, MSWEP and TERRA can be higher than IMD by up to 19%, 37%, 10% and 6%, respectively. The discrepancy between these datasets and IMD can even be larger in certain river basins.

## 5    Discussion

The following is a discussion on the limitations of this study, followed by a discussion on some spurious issues with the IMD dataset which were encountered during the course of this study. Some ideas on how to better quantify $UoP$ are also discussed.

### 5.1    Limitations

The watersheds affected by $UoP$ were identified by analyzing the extent of annual water imbalance. As such, the results are dependent on the reliability of the data and strength of the assumptions used within the analysis. The data used here not only includes hydrometeorological data, such as observed streamflow ($R$) and estimates of $ET$, but also catchment boundaries and information on water management. The limitations of these datasets and also the limitations imposed by the assumptions used within this analysis are discussed here.

#### 5.1.1    Limitations with data

The GHI dataset (Goteti, 2023) was chosen in this study because of the quality-controlled nature of catchment boundaries and $R$ data used in its development. GHI stations used in this study were those that had a catchment area discrepancy of less than 5% when compared with CWC. It is assumed that catchment boundaries used here are reasonably accurate, and any errors with such boundaries and are not likely to cause the water imbalance identified in this study.

As mentioned earlier in Section 2.1, GHI is limited to Peninsular India, and $R$ data for Northern India was compiled from CWC-19 (2019). Such annual and monthly $R$ data were compiled from daily records which are known to have missing days. Hence, the actual $R$ is very likely higher than observed $R$. Thus, it is expected that there would be more imbalanced station-

years. Moreover, as additional $R$ data from other gauging stations becomes available, particularly in the mountainous portions

of Northern India, many other watersheds affected by $UoP$ would be identified. All of the $R$ data used here is directly or indirectly through the CWC. Legacy data from other sources is also available but is not considered reliable (see Goteti, 2023). Studies have reported that $R$ based on rating curves could have significant errors (e.g., Di Baldassarre and Montanari, 2009; Kiang et al., 2018). Huang et al. (2023) estimate that about 70% of the global streamflow gauging stations analyzed in their study have a bias in catchment discharge of greater than 10%. None of the stations from Huang et al. (2023) are present within

this study's analysis domain. It is not known to what extent $R$ from CWC is derived from rating curves, or to what extent such data is affected by measurement or other errors, or to what extent such errors in daily or sub-daily streamflow affect the annual or monthly streamflow data used in this analysis.

ET estimates from GLEAM were used in this study instead of those from NTSG. While there is reasonable correlation between the two products, GLEAM-based $ET$ is generally lower than NTSG-based $ET$ (see Appendix C). Goroshi et al.

(2017) indicated that NTSG underestimates lysimeter-based $ET$ observations across many locations in India. This would imply that GLEAM would further underestimate such $ET$ observations. Thus, the $ET$ values from GLEAM used in this analysis should be considered a lower bound for $ET$. If more accurate $ET$ estimates were to become available, it is expected that there would be more imbalanced station-years.

Management of water, such as groundwater extraction, reservoir storage and water diversion (imports or exports), is shown

to be relatively minimal in the imbalanced watersheds (relative to annual $P$). Extent of groundwater extraction is available at a district resolution and only for select years. Some studies have urged caution when interpreting trends in groundwater levels from CGWB (e.g., Hora et al., 2019). The quantification of groundwater storage in the study domain is particularly challenging due to varying geological settings (alluvial versus hard rock aquifers), extensive and unregulated withdrawal for irrigation use, and changing energy policies (Panda et al., 2022). Dams considered here were only from India and include only the large

dams available via the NRLD inventory. It is possible that smaller, or other, dams present in the watershed, and not included within NRLD, could be causing some of the water imbalance. The effect of such dams at an annual timescale is assumed to be minimal. Data on water diversions is available only for select sub-basins of the major basins of India. Ideally, watershed-scale management information is needed to reliably conclude if management is the cause of water imbalance in these imbalanced watersheds. There are also a number of local watershed development projects being pursued in the forested and mountainous

regions of India, such as those reported by Chauhan (2010). The effect of such development on the hydrologic budget of the watersheds analyzed here is unknown.

GRACE-based annual changes in $\Delta TWS$ are useful in understanding the effect of such changes on the annual water budget. As discussed by Humphrey et al. (2023), numerous assumptions went into the processing of raw GRACE data and one has to exercise caution when interpreting the end products derived from raw data. The effective resolution of GRACE is about 300

km x 300 km. Thus, watershed-scale annual $\Delta TWS$ values for the imbalanced watersheds in Figure 11 are representative of coarser-scale patterns, and complement the data on watershed management summarized in Figure 5. GRACE-based $\Delta TWS$ cannot be directly compared to changes in local water table. Some recent studies have assimilated GRACE observations into

hydrological models to better capture finer-scale groundwater storage changes (Li et al., 2019). The use of $\Delta TWS$ from such studies could be explored in future work.

The accuracy of gauge-based products such as IMD is dependent on the underlying gauge data as well as the specific interpolation procedures used to create the gridded data. If raw gauge data is available, it might be worthwhile to compare such data with the other $P$ datasets for select high-intensity storms in the imbalanced watersheds. The reader is referred to the studies of Prakash et al. (2015a) and Prakash et al. (2019) for information on how such comparisons could be made and also on the challenges involved in creating gridded products. The $P$ datasets analyzed here are continuously going through revisions and

improvements. As such, the results presented here are relevant to the specific versions of the datasets analyzed here, and should not be considered applicable to future versions of such products. However, the generic nature of the methodology adopted here would still be applicable to such products.

### 5.1.2    Limitations with the methodology

Watershed boundaries do not always coincide with underlying aquifer boundaries (e.g., Liu et al., 2020), and so one cannot

always assume that water flowing out of a watershed is completely generated within the watershed. A number of studies have discussed the important role of inter-watershed groundwater flow (IGF) (e.g., Fan, 2019), including within high mountain environments such as the Brahmaputra Basin (e.g., Somers and McKenzie, 2020; Yao et al., 2021). The contribution of IGF to streamflow depends on a number of factors such as geology, topography and climate, among others. While some studies have identified that karst aquifers are present in select parts of India (Dar et al., 2014), extensive watershed-specific hydrogeologic

field investigations are needed (such as those by Yao et al., 2021) to quantify the effect of IGF on streamflow. This study assumes IGF to be negligible. It is possible that some or all of the watersheds analyzed in this study are affected by IGF. However, one should not assume that all instances of observed annual water imbalance are solely due to IGF. As discussed in Appendix F, there are watersheds within the study domain where it appears that IGF is unlikely the cause of the observed water imbalance.

Yao et al. (2021) analyzed the contribution of groundwater to $R$ in the upper reaches of the Brahmaputra Basin. Several watersheds from the study of Yao et al. (2021) had annual runoff coefficients greater than 1.0 due to contribution from IGF as well as snowmelt and permafrost thawing. Contribution of seasonal snow melt is implicitly considered within observed $R$, but glacier melt has not been considered. In the Himalayan mountains, glacier melt could sometimes be a significant portion of the annual runoff and could even exceed snow melt (e.g., Mukhopadhyay and Khan, 2015). In such watersheds where glacier melt

is substantial, annual observed $R$ could be higher than annual $P$, despite there being no management. GRACE-based $\Delta TWS$ is supposed to capture storage changes due to glacier melt at the spatial scale of major river basins, but not across smaller watersheds. It is possible that the approach adopted in this study could incorrectly identify such watersheds to be affected by $UoP$.

     The identification of watersheds affected by $UoP$ focuses on those regions where there is relatively minimal effect of

management or where annual $\Delta TWS$ is minimal relative to $P$. It is not clear how to identify $UoP$ when there is moderate to extensive management or where annual $\Delta TWS$ is substantial relative to $P$. Analyzing the relative magnitudes of the individual

**Table 6.** Trend in annual basin-aggregated $P$ (mm/year) for WY 1985-2014 for select datasets and select river basins. Statistically significant values at the 95% confidence level are indicated by '(*)'.

| Basin | IMD | ERA5 | IMDAA | MSWEP | TERRA |
|---|---|---|---|---|---|
| All India | -1.7 | +3.2 | +3.6 | +2.1 | +2.3 |
| Barak | -33.2(*) | -11.3 | -5.9 | -16.4(*) | -12.2 |
| Brahmaputra | -19(*) | -19.1(*) | -13.3 | -13.9(*) | -7.5 |
| Ganga | -4 | +1.7 | +0.8 | +2.6 | -4.1 |
| Indus | -12.2(*) | -6(*) | -4.4 | -3.2 | -1.2 |
| WFR North | +8.6 | +17(*) | +22.7(*) | +17.3(*) | +36.6(*) |
| WFR South | +23.7(*) | +21.4(*) | +18.9(*) | +18.4(*) | +16.3 |

terms of the water budget might not be the way to identify $UoP$ under such circumstances. The two water imbalance scenarios investigated in this study (Eq. 6 and Eq. 7) are only two of the many possible scenarios. The imbalanced watersheds identified in this study are dependent on the formulation of such scenarios (see Sections 4.1 and 4.3).

The formulation of Scenarios I and II (Eq. 6 and Eq. 7) relies more on $R$ and less on $ET$. This is because while observations of $R$ are available, observed $ET$ at the scale of the watersheds analyzed in this study is non-existent. Satellite-based $ET$ was used as a proxy for observed $ET$. However, such $ET$ data can have substantial biases (e.g., Goroshi et al., 2017; Goteti, 2022). Hence, observed $R$ is assumed to be more reliable than satellite-based $ET$. If one had more reliable estimates of $ET$, then the formulation of the scenarios could be revised to include other instances of spurious water imbalance.

## 540  5.2  Spurious patterns within IMD

During the course of this analysis, several potential issues with trends in the IMD dataset were encountered. The following is a discussion on basin-scale trends in the IMD dataset and those present in other datasets. For the purposes of this discussion, the spatial domain is limited to the political boundaries of India where IMD data is available. Basin-scale aggregation of gridded $P$ was performed only using the grids falling within India's boundaries.

Trends in the four datasets identified in Section 4.3 are compared against those in IMD. Trends in basin-aggregated annual $P$ for WY 1985-2014 were estimated using the non-parametric Thiel-Sen slope (Helsel et al., 2020) making use of the R statistical package 'RobustLinearReg' (Hurtado, 2023). Table 6 shows the trends for select basins where mountains are present. Table S2 (Supplement) shows the trends for all of the basins and all of the $P$ datasets.

Annual $P$ from IMD for the whole of India shows a decreasing trend of -1.7 mm/year. In contrast to IMD's decreasing trend,
all other datasets have an increasing trend. However, none of these trends are statistically significant at the 95% confidence level. There is substantial variation in regional trends as is evident in the trends presented for individual basins. For the IMD dataset, Barak, Brahmaputra, Ganga and Indus basins in Northern India show decreasing trends. However, other datasets do not always have the same magnitude or sign as IMD. For instance, for the Ganga Basin, IMD shows a negative trend of -4

mm/year, while ERA5, IMDAA and MSWEP show a positive trend. None of these trends are statistically significant at the
95% confidence level. For the wettest basin of India - the WFR South Basin, all datasets show a positive trend with most of
them being statistically significant. Based on Table S2 (Supplement), there appears to be more consistency in trends between
IMD and other datasets for the basins of Peninsular India compared to the basins of Northern India.

Another issue which was encountered during the course of this analysis was abrupt changes in the time series of $P$ from the
IMD dataset, particularly in the earlier part of its record. The time period of interest here is the 20-year period of WY 1981-
2000 relative to the prior 20-year period of WY 1961-1980. Time series of basin-averaged annual $P$ from IMD is compared
with the corresponding time series from three $P$ datasets which have data available during these periods - APHRO, ERA5 and
TERRA. The reader should note that while ERA5 and TERRA are reanalysis-based datasets, APHRO is a gauge-based dataset
similar to IMD.

Figure 13 shows such comparisons for the Barak and Indus basins, while the Supplement (Figure S17 through S39) shows a
similar time series comparison for all of the major basins. Both annual (thin lines) and 9-year running average (thick lines) are
shown in Figure 13 to highlight the short and long-term changes in $P$ in each of the datasets. Figure 11 shows that for the Barak
Basin, IMD shows an increase in average annual $P$ of about 22% for WY 1981-2000 relative to WY 1961-1980. However,
APHRO, ERA5 and TERRA show a change of 8%, -3% and 5%, respectively. Also, IMD has a distinct visual increasing trend
from low values in the early 1960s to high values in the early 1990s. Such a pattern is not present in APHRO, ERA5 or TERRA.
Similarly, for the Indus Basin, IMD shows an increase in average annual $P$ of about 35% for WY 1981-2000 relative to WY
1961-1980. However, APHRO, ERA5 and TERRA show a change of 5%, 4% and 3%, respectively. IMD, once again, has a
distinct visual increasing trend from low values in the mid 1970s to high values in the late 1990s. Such a pattern is not present
in APHRO, ERA5 or TERRA.

Overall, the above discussion highlights two related issues with the IMD dataset. First, trends present within the IMD dataset
are not always present in other datasets. Second, conspicuous temporal shifts present in the IMD dataset are not present in other
datasets. Lin and Huybers (2019) noted a potentially spurious shift in IMD's dataset over Central India. It is not known if, and
to what extent, such issues are caused by $UoP$ (or overestimation of $P$) within these datasets.

## 5.3 Interim Measures

Solving the problem of $UoP$ either by increasing the station density in relevant areas, or by monitoring and analyzing extreme
$P$ events and rainfall-runoff relationships for such events, or by any other means requires significant planning and resources
from the relevant government agencies. Such efforts are strongly encouraged by the authors of this paper. In the interim, there
are several useful and feasible ideas the community could pursue to help address the issue of $UoP$. Following is a discussion
of such ideas.

Raw station data from IMD would be extremely helpful in resolving discrepancies associated with trends and discrepancies
with other datasets. However, such data is not publicly available. IMD could help the scientific community better quantify $UoP$
by making such raw station data publicly available. Within its gridded product, IMD could include the number of reporting
stations present in each grid, and the number of neighboring stations used in estimating the value at a particular grid. Such in-

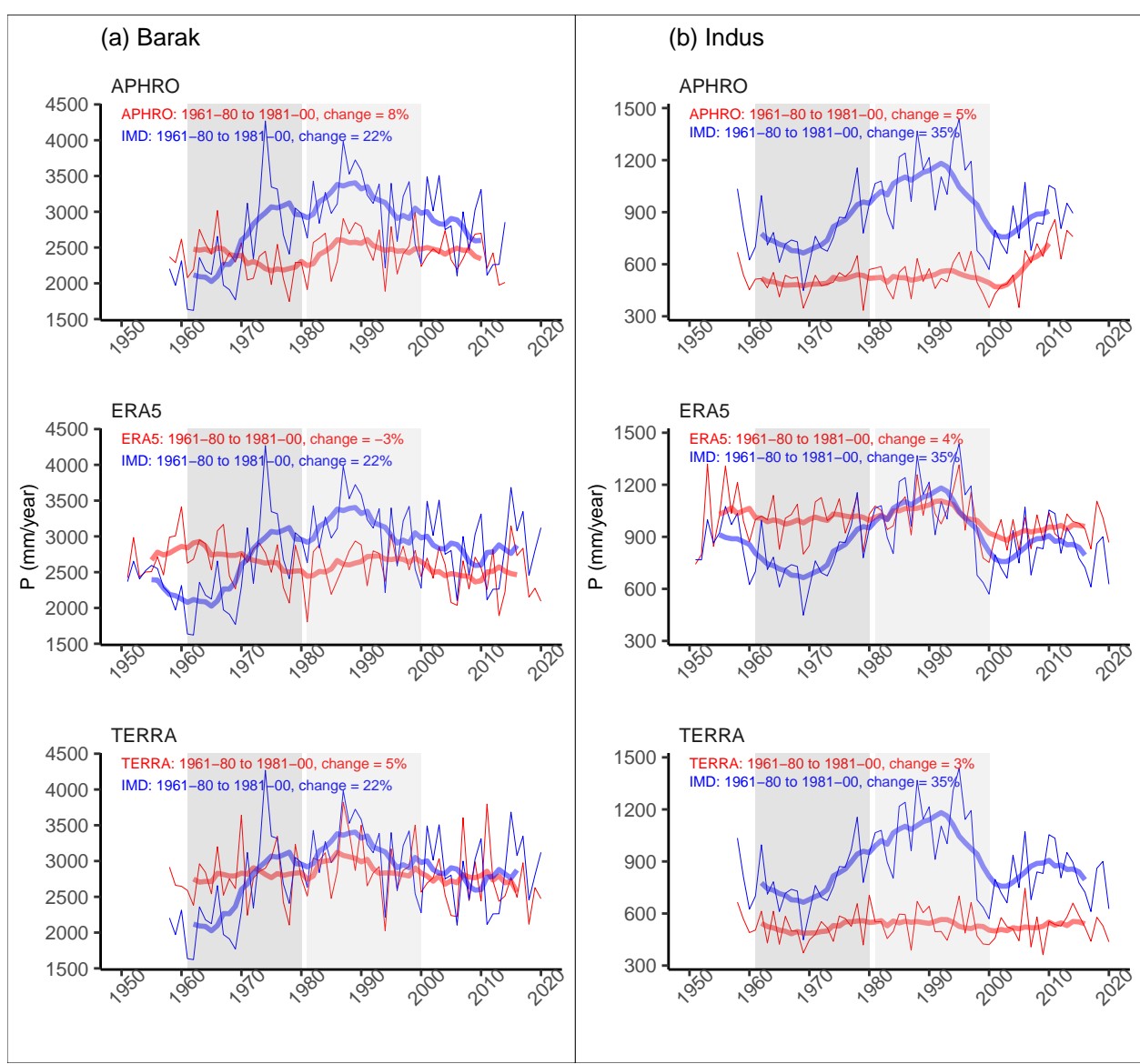

**Figure 13.** (a) Annual $P$ for Barak Basin for select datasets. IMD (blue) is compared against select datasets. The periods of interest, WY 1961-1980 and WY 1981-2000, are highlighted. Thin lines show the annual values while the thick lines show the 9-year running average. (b) same as (a) but for the Indus Basin.

formation would help understand the difference between actual and operational (or reported) station density, historical changes in such station densities, and also identify any spurious gridded values associated with extreme $P$ events.

Other data from India's water agencies would also be valuable in addressing $UoP$. Currently, availability of observed $R$ data is limited to Peninsular India. Most data for Northern India is 'classified' by CWC, and annual data is available for a limited number of stations within reports published by CWC. Daily observations of $R$ are useful in understanding rainfall-runoff relationships for extreme $P$ events, while monthly and annual data are useful in identifying gross $UoP$.

     Some modeling studies have demonstrated the ability of high resolution simulation models to capture $P$ in watersheds domi-
nated by hilly or mountainous terrain. For instance, Li et al. (2017) implemented the Weather Research and Forecasting (WRF) Hydro model in a high-resolution setting (3 km grid) across a mountainous watershed of Northern India. They demonstrated that such a system can reasonably simulate $P$ and can overcome the deficiencies of typical gauge-based products and satellite-based products. Hunt and Menon (2020) also used the WRF-Hydro modeling system in a high resolution setting (4 km grid) to analyze $P$ during the catastrophic flooding of 2018 in the State of Kerala in Peninsular India. Their study was also able
to reasonably capture the spatial structure and magnitude of observed $P$. Such modeling studies should be pursued further to better identify and quantify $UoP$ within traditional products.

     Until the issue of $UoP$ is resolved, analysts should exercise caution when analyzing trends based on IMD's gridded data alone, or when evaluating other products against IMD, such as reanalysis or satellite-based products. Also, IMD should not be considered the 'true' benchmark in such evaluations. Studies using IMD to analyze water budgets should exercise caution
when interpreting their results, particularly in watersheds or regions where $UoP$ is identified to be relatively high. Some of the reanalysis-based datasets, such as ERA5 and IMDAA, suffer from $UoP$ to a much lesser extent than IMD. It is not known to what extent such datasets are affected by overestimation of $P$. Such datasets should be evaluated further to understand their relative merits and demerits.

## 6   Conclusions

This study analyzed gross underestimation of precipitation ($UoP$) in India using a water balance approach across 242 watersheds of Northern and Peninsular India. Gross $UoP$ was identified by comparing water year (WY) based volume of observed annual $P$ against observed annual streamflow ($R$), and $P$ against the sum of $R$ and satellite-based evapotranspiration ($ET$). Across many watersheds of both Northern and Peninsular India, the spurious water imbalance scenarios of $P \leq R$ or $P << R + ET$ were realized. It is shown that the occurrence of such imbalance is unlikely due to large-scale management of
water, such as groundwater extraction, reservoir storage and diversions. It is also shown that the occurrence of such imbalance is unlikely due to annual changes in terrestrial water storage. Assuming data on $R$ and $ET$ to be reliable, it is concluded that $UoP$ is the likely cause of such spurious imbalance. The effect of inter-watershed groundwater flow has been ignored in this study. However, it appears that such groundwater flow is unlikely the cause of spurious water imbalance observed in some of the watersheds.

620 All 12 state-of-the-art $P$ products analyzed here suffer from $UoP$, but to varying extent. Within the often used IMD dataset, $UoP$ is an issue in most river basins of India and is present throughout the historical record, including the decade of 2010s. Based on the limited observation data available, $UoP$ is found typically in the relatively wet regions of India. Thus, our understanding of the hydrology of India is limited by inadequate $P$ data, particularly in these wet regions, some of which have experienced catastrophic flooding during the recent years. Moreover, the $P$ product from IMD, which is typically the

625 benchmark in many hydrological and environmental studies across India, suffers from $UoP$ more than some products based on reanalysis. The $P$ from such products tends to be much higher than IMD across most river basins of India. Furthermore, such products do not have the spurious temporal patterns found in IMD. Studies using the IMD dataset should exercise caution, particularly in the regions with hilly or mountainous terrain. This study not only highlights a major limitation of existing $P$ products over India but also other data-related obstacles faced by the research community.

630 *Data availability.* Summary graphics and tables discussed in Appendices A, B, C, D, E and F are included in the Supplement. Information on gauging stations used in this analysis, including station names and locations, GIS data on their catchment boundaries, and GIS data on river basin boundaries is included with the Supplement. Also included is a plain text file on the time series of hydrometeorological data associated with each station. This study only uses publicly available data sources and they were cited wherever applicable.

## Appendix A: PBCOR dataset

A bias-correction factor (CF) is defined the ratio of actual $P$ to observed $P$. The PBCOR dataset (Beck et al., 2020) provides average annual and monthly estimates of bias-corrected $P$ at a resolution of 0.05 deg. If PBCOR's estimates were to be considered reasonable, then the ratio of PBCOR to IMD's observed data represents the CF associated with IMD data. Such estimated CFs are presented in Figure 1 and in Figure S1 of the Supplement.

Estimates from PBCOR are specific to one of three reference climatology datasets used within its development - CHELSA V1.2, CHPclim V1 and WorldClim V2. The time period corresponding to each of these reference climatologies is as follows: for CHELSA V1.2 it is 1979-2013, for CHPclim V1 it is 1980-2009, and for WorldClim V2 it is 1970-2000. PBCOR was aggregated from its native 0.05 deg resolution to the IMD resolution of 0.25 deg by appropriately accounting for spatial overlap between the 0.05 deg and 0.25 deg grid meshes. Next, the long-term average monthly $P$ was estimated for the IMD grids, for each of the above time periods corresponding to each climatology. The ratio of aggregated PBCOR data at 0.25 deg and IMD data, as shown in Eq A1 - A3, is the estimated CF and is shown in Figure S1 of the Supplement. The CFs shown in Figure 1 are specific to CHPclim.

$$CF^{CHELSA} = \frac{\overline{P}^{CHELSA}}{\overline{P}^{IMD}_{1979-2013}} \tag{A1}$$

$$CF^{CHPclim} = \frac{\overline{P}^{CHPclim}}{\overline{P}^{IMD}_{1980-2009}} \tag{A2}$$

$$CF^{WorldClim} = \frac{\overline{P}^{WorldClim}}{\overline{P}^{IMD}_{1970-2000}} \tag{A3}$$

## Appendix B: Precipitation

Additional graphics and tables on $P$ datasets are presented in the Supplement, including: (1) maps of average annual $P$ for WY 2007-2014; (2) table showing basin-scale ratio of $P$ from each product against IMD for WY 2007-2014; (3) maps of monthly total $P$ corresponding to three flooding events - Assam flooding of June 2012, Jammu and Kashmir flooding of September 2014 and Kerala flooding of August 2018; and (4) time series of basin-averaged $P$ for each product compared against IMD, and trends associated with such time series. Following is a brief description of these graphics and tables.

For each of the 12 $P$ products analyzed in this study, annual average $P$ for the common eight year period of WY 2007-2014 is presented (Figures S2-S13). It can be seen that all of the products have the broad spatial pattern of relatively wet Western Coast, Northeastern India and Northernmost India. The relatively dry regions of Northwestern India and interior Peninsular India are also evident in these maps.

The ratio of basin-averaged annual $P$ between each product and IMD is presented in Table S1. The same common data period of WY 2007-2014 is used for estimating these ratios. A ratio greater (smaller) than 1.0 indicates that $P$ from the particular product is higher (lower) than IMD. Across the whole of India, ERA5 and IMDAA have the highest ratios (1.15 and 1.31, respectively), while APHRO and GSMAP have the lowest ratios (0.91 and 0.90, respectively). In general, ERA5 and IMDAA have values greater than 1.0 for most river basins in India.

Spatial maps of monthly gridded $P$ from the various datasets for select major flooding events from the recent past are also presented. The events discussed here occurred in regions with mountainous terrain within the past 10 years and were identified from the Dartmouth Flood Observatory's global archive of major flood events (Brakenridge, 2023). These events include the flooding in the State of Assam (Northeastern India) in June 2012, flooding in the State of Jammu and Kashmir (Northernmost India) in September 2014, and flooding in the State of Kerala (Southwestern India) in August 2018. For the Assam floods of 670    2012 (Figure S14), the heavy $P$ cluster of greater than 1500 $\mathrm{mm/month}$ present in IMD is also present in ERA5 and IMDAA. While the rest of the datasets have such a cluster to a smaller extent, CHIRPS and GSMAP do not have such a cluster. For the Jammu and Kashmir floods of 2014 (Figure S15), the heavy $P$ cluster of 400-800 or greater than 800 $\mathrm{mm/month}$ present in IMD is also present in several of the datasets. In some datasets, such as CHIRPS, IMERG and TERRA, the cluster is larger and contains $P$ of higher magnitude than IMD. GSMAP, PERSIANN and SM2RAIN do not have such a cluster. For the Kerala 675    floods of 2018 (Figure S16), the heavy $P$ cluster of 1000-1500 or greater than 1500 $\mathrm{mm/month}$ present in IMD is present only to a limited extent in IMDAA but absent in the rest of the datasets. While these maps correspond to unusually wet events, it is evident that there are substantial differences between the different datasets for such events.

Time series of area-averaged annual $P$ from IMD is compared with the corresponding time series from other datasets in Figures S17-S39. For the purposes of this comparison, all datasets were limited to the boundaries of India since IMD is limited 680    to such boundaries (i.e., IMD-APHRO is not used here). Basin-scale aggregation of gridded $P$ was performed only using the grids falling within India's boundaries. The 9-year running average is also shown for each of the datasets to highlight the temporal trends in each of the datasets. The trends presented are based only on the period WY 1985-2014, if data was available.

**Appendix C: ET, GLEAM vs NTSG**

As discussed earlier, GLEAM and NTSG datasets were considered for this analysis, but GLEAM dataset was used because of 685    the longer time span of this dataset and its availability to the present time. $ET$ from the Numerical Terradynamic Simulation Group (NTSG) at the University of Montana (Zhang et al., 2010) provides estimates of monthly $ET$. Goroshi et al. (2017) compared NTSG estimates with lysimeter-based $ET$ observations across many locations in India, and found that while there is reasonable agreement between them at seasonal and annual timescales, NTSG was found to underestimate observed $ET$ during the monsoon (June-August) and post-monsoon (September-November) seasons. Goteti (2022) noted a similar issue 690    with GLEAM in Godavari and Krishna basins of Peninsular India.

A comparison of GLEAM and NTSG, for the overlap period of WY 1982-2012 is presented in the Supplement (Figure S40). The basin-aggregated average $ET$ for the major basins is shown. For ease of visualization, the extreme values - the lowest and

highest annual values within each basin, corresponding to the NTSG dataset were excluded. In general, there is a reasonable correlation between GLEAM and NTSG across many basins. GLEAM values are also lower than NTSG for many basins, as indicated by the negative values of percent bias ('pbias'). Given NTSG's low bias, this indicates an even larger low bias with GLEAM's $ET$.

## Appendix D: Water Management

### D1 Groundwater Extraction

Groundwater extraction and recharge estimates are available from India's Central Ground Water Board (CGWB, http://cgwb.gov.in/) for select years (https://ingres.iith.ac.in/). These estimates are available at the administrative district resolution. Districts in India are the third administrative tier, following national and state tiers. However, official GIS data on district boundaries is not readily available from Indian agencies. Moreover, administrative boundaries, including district and state boundaries, have been subject to change in the recent past (e.g., the states of Andhra Pradesh and Telangana). Available GIS data on district boundaries from geoBoundaries (Runfola et al., 2020) was compiled so that the names of the districts and their areas closely matched those from CGWB. The estimates from two recent years, WY 2019-20 and WY 2021-22 are used here. In order to have reliable district boundaries and also retain as much information as possible, the data for the State of Andhra Pradesh was taken from WY 2019-20 and data for the remaining states was from WY 2021-22.

Total groundwater extracted was estimated by CGWB for three different categories: command (or irrigated) - 'C', non-command (or non irrigated) - 'NC', and poor quality - 'PQ'. Consistent with CGWB, only the 'C' and 'NC' categories were used here to estimate the total volume of groundwater extracted. Annual $P$ is also available from CGWB, and it was converted to a volume using the area of each district estimated by CGWB. Finally, the district-wise extent of annual groundwater extraction is quantified as a fraction of the annual $P$ and is presented in Figure 5.

### D2 Imports and Exports

CWC-19 (2019) estimated the water resources availability for the major river basins of India and their sub-basins for WY 1985-2014, and during this process quantified the various inputs and outputs to these basins, including both natural and human-caused factors (Annexures A-S). The variables quantified by CWC-19 include volume of basin-aggregated $P$, imports to the basin, and exports from the basin. Both imports and exports are expressed as a fraction of annual $P$, and the maximum value of such estimates is presented in Figure 5.

### D3 Dams and Reservoirs

Information on large dams in India was obtained from the National Registry of Large Dams (NRLD, 2019). Raw data from NRLD for 5,745 large dams was available as a Portable Document Format (PDF) file. This data was first compiled into a spreadsheet and then imported into a GIS software to perform basic quality checks. Locations of dams with missing latitude or

longitude, or those falling in the ocean or outside of India's political boundaries, were deemed spurious and discarded. Thus, a total of 5,629 dams were considered for this analysis. The specific purpose of the dam - such as storage, irrigation or hydro power, was not considered. For ease of illustration, the locations of the dams were mapped to a 25 km grid and the number of dams per each such grid is displayed in Figure 5. It is evident that the density of dams is the largest in the arid Western India. The density of dams is low along the Western Coast of India, in the Gangetic Plains and the mountainous portions of Northern India.

Other than the coordinates of the dam, the attributes of interest for each dam are the year of construction of the dam, the maximum live storage capacity and the total storage capacity of the dam. While a vast majority of the dams had such information available, some dams had this information missing. If the year of construction of a dam was missing, it was assumed to be 1950 - the earliest year of analysis. If live storage capacity was missing, it was assumed to be 0.9 times (or 90%) of the total storage capacity. The factor of 0.9 used here was based on the median ratio of such a factor where information was available. Once all the relevant information on dams was compiled, the river basin and watershed associated within each dam was identified using a GIS analysis. Thus, for a given streamflow gauging station, all the dams present in the upstream catchment area were identified. The annual cumulative live storage capacity for each gauging station, and for each year, was estimated as the sum total of all such upstream dams, taking into account the year of construction of the dam. In Figure 5, cumulative live storage capacity in WY 2019 (the latest year for which data is available from NRLD) is expressed as a fraction of average annual $P$.

## Appendix E: Estimates of $\Delta TWS$

Temporal changes in the Earth's gravity field measured by the Gravity Recovery and Climate Experiment (GRACE) satellite mission (Tapley et al., 2004) have been used to infer changes in total terrestrial water storage ($TWS$) (Rodell et al., 2009). GRACE does not provide the total amount of $TWS$ nor its long-term average ($\overline{TWS}$), but instead provides estimates of $TWS$ anomalies (i.e., $TWSA = TWS - \overline{TWS}$) (Humphrey et al., 2023). Raw data from GRACE can be processed using several mathematical techniques to generate useful end products, and there are many such products currently available (Humphrey et al., 2023). This study uses the often-used $0.25 \deg$ ($\sim$25 km) resolution anomalies from the Center for Space Research (Save et al., 2016; Save, 2020).

The difference between the anomalies at two different times gives an estimate of the change in $TWS$ (or $\Delta TWS$) over that time period. Change in annual $TWS$ (or $\Delta TWS$) is of primary interest for the purposes of this analysis. Considering the definition of WY used in this study (period of June through May, see Section 1.1), $\Delta TWS$ was estimated using an equation similar to Eq. E1. As discussed by Humphrey et al. (2023), equations Eq. E1 is an approximation since GRACE monthly anomalies do not correspond to exact calendar months.

$$\Delta TWS_{WY\ 2010} = TWSA_{May\ 2011} - TWSA_{June\ 2010} \tag{E1}$$

**Table F1.** Annual $P$, $R$ and $ET$ (in MCM) for WY 2007 for the select watersheds in Fig. F1.

| Basin | Site | Area (km²) | P (IMD) | R | ET | R / P | (R + ET) / P |
|-------|------|-----------|---------|-----|-----|-------|--------------|
| Cauvery | Kudige | 1,742.8 | 3,794 | 3,932 | 1,962 | 1.04 | 1.55 |
| Cauvery | Sakleshpur | 615.6 | 1,572 | 2,018 | 681 | 1.28 | 1.72 |
| Krishna | Shimoga | 2,761.3 | 8,197 | 7,303 | 3,078 | 0.89 | 1.27 |
| WFR South | Addoor | 718.7 | 3,217 | 2,612 | 1,017 | 0.81 | 1.13 |
| WFR South | Avershe | 299.6 | 1,033 | 1,381 | 398 | 1.34 | 1.72 |
| WFR South | Bantwal | 3,295.4 | 13,058 | 14,952 | 4,432 | 1.15 | 1.48 |
| WFR South | Erinjipuzha | 912.0 | 3,389 | 3,031 | 1,244 | 0.89 | 1.26 |
| WFR South | Halady | 566.4 | 2,646 | 2,410 | 726 | 0.91 | 1.19 |
| WFR South | Yennehole | 356.9 | 1,527 | 1,535 | 510 | 1.01 | 1.34 |

Consistent with the other analyses of this study where hydrological variables were represented as a fraction of annual $P$ (Section 2.4), $\Delta TWS$ was also estimated as a fraction of annual $P$, using $P$ from the IMD-APHRO dataset (Section 2.2). The annual maps of $\Delta TWS$ are shown in the Supplement (Section S6), for years when GRACE data is available for the months of May and June (starting WY 2002), and for years for which the IMD-APHRO dataset is available (up to WY 2014). Overall, there are 10 WYs during WY 2002-2014 for which such maps could be created. Figure 6 shows the grid-wise maximum and minimum $\Delta TWS$ across all such WYs.

## Appendix F: Case Study

In order to demonstrate that inter-watershed groundwater flow (IGF) does not always explain the observed water budget imbalance, a case study is presented here. Annual and monthly water balance of nine select watersheds in Southwestern India are examined (Fig. F1). WY 2007 is chosen as an example, and it is one the many years where watershed imbalances were observed. The selected watersheds have been identified in this study to be experiencing $UoP$ using the IMD dataset (Section 4.1). These watersheds are part of three major basins (Cauvery, Krishna and WFR South) and their catchment areas range from about 300 to 3,300 km² (Table F1). The combined area of these watersheds is about 11,270 km². These watersheds are adjacent to each other, and relative to the surrounding terrain they are located at a higher elevation (Fig. F1(c)).

Table F1 shows the annual $P$, $R$ and $ET$ for WY 2007 for the nine watersheds. It is evident that the annual runoff coefficient ($R/P$) is at least 0.81, and exceeds 1.0 for five of the nine watersheds. The ratio of $(R + ET)/P$ ranges from 1.13 to 1.72. Monthly values of $P$, $R$ and $ET$ for the monsoon season (June to September) during WY 2007 are shown in Figure F2. Except for the smaller watershed of Halady, $R$ exceeds $P$ for multiple months during the monsoon season. Thus, these watersheds are simultaneously experiencing the water imbalance scenarios of $P \leq R$ or $P << R + ET$. The temporal pattern of $P$ and $R$ is generally similar across the watersheds - with July being the month of largest $P$ and $R$.

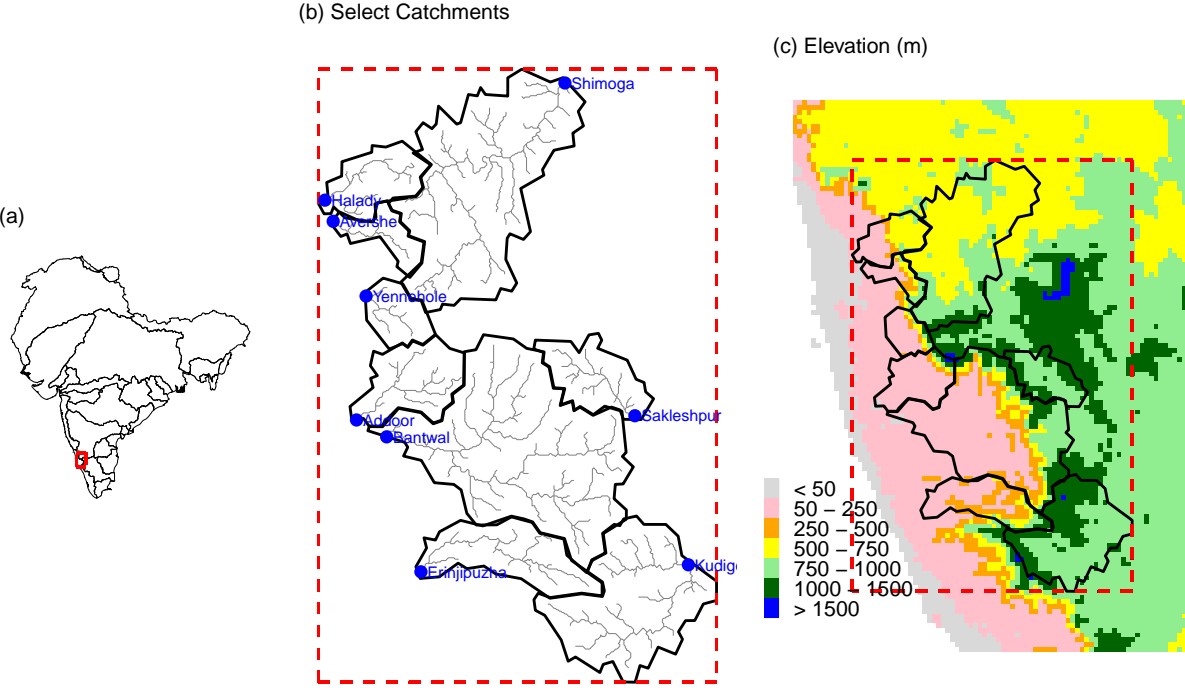

(b) Select Catchments

(c) Elevation (m)

(a)

< 50
50 − 250
250 − 500
500 − 750
750 − 1000
1000 − 1500
> 1500

**Figure F1.** Select watersheds analyzed in this case study. (a) Red box shows the location of the nine watersheds within the study domain; (b) streamflow gauging stations (blue dots) at the outlets of the watersheds (black lines) and the river network (grey lines); (c) topography of the select watersheds and the surrounding region.

It is not known if, and to what extent, IGF is present within these watersheds. The possibility of IGF causing the observed
spuriously high runoff coefficients is examined here. If IGF is present in this region, groundwater can flow between these watersheds and (or) can flow into these watersheds from the adjacent terrain.

In case of flow between the watersheds, some of them would be losing streamflow, while the others would be gaining streamflow. Under such circumstances, some watersheds would have lower runoff coefficients while others would have higher coefficients. One would not see the simultaneous occurrence of monthly runoff coefficients greater than 1.0 across most of the
780 watersheds, such as those during the months of July and August (Figure F2). This simultaneous occurrence of high monthly runoff coefficients also happens in other years (but is not shown here). Thus, it appears that IGF between the watersheds is unlikely the cause of observed watershed imbalance.

IGF from the surrounding terrain into these watersheds is unlikely since these watersheds are at a relatively higher elevation (Fig. F1(c)). Groundwater from lower elevations would have to flow against gravity in order to reach these watersheds at higher
elevations. Thus, it appears that IGF from the surrounding terrain is unlikely the cause of observed watershed imbalance.

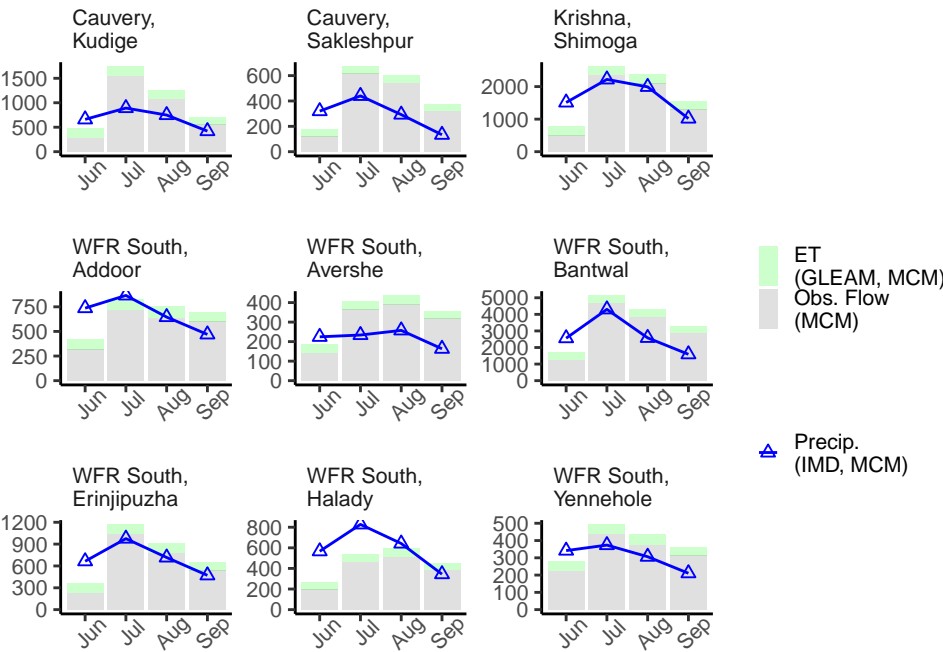

**Figure F2.** Monthly $P$, $ET$ and $R$ (in MCM) for WY 2007 for the select watersheds in Fig. F1. Only the months of the monsoon season, June-September, are shown.

Based on the above discussion, it is reasonable to say that IGF is not the predominant cause of the observed water imbalance in the above watersheds. However, the reader should note that IGF can still be present. Only field data can truly reveal the pattern of groundwater flow occurring under these watersheds and the extent of its contribution to streamflow. The above analysis was not performed in other parts of the study domain because of data limitations - inadequate monthly streamflow and the non-contiguous nature of imbalanced watersheds.

*Author contributions.* GG and JF collaborated on the conceptual framework of the analysis and the writing of the paper. GG performed the analyses.

*Competing interests.* No competing interests are present.

*Acknowledgements.* A number of publicly available datasets were used in this study and were cited wherever applicable. Software used in this study includes the R statistical computing and graphics software for data analysis (https://www.r-project.org/) and QGIS for GIS analysis

(https://qgis.org/en/site/). Political boundaries for India were obtained from the Survey of India (https://surveyofindia.gov.in/pages/outline-maps-of-india).

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
