# Peer review of "Extent of gross underestimation of precipitation in India"

_Hydrology and Earth System Sciences, 2024_

## Referee Comment (RC2)

[referee-annotated manuscript omitted]

---

## Author Comment (AC1)

**Response to Reviewer #1's comments:**

The manuscript entitled "Extent of gross underestimation of precipitation in India" by Goteti and Famiglietti, analyzes watershed-scale underestimation of precipitation over India using various precipitation datasets within a water imbalance analysis. The authors have designed the study with relevant methodology through the knowledge attained from associated literature. There are some corrections required in the manuscript, although the authors have explained the results and the methodology in detail, there are a few disconnections between sentences and minor grammar corrections required. I have raised a few questions that require clarification, also suggested some necessary modifications and additions to the manuscript.

The authors would like to thank the reviewer for the feedback and the suggestions made to improve the manuscript. The authors generally agree with the comments, and revisions were made accordingly. Please see below the specific responses to the individual comments.

**Summary of major changes** (based on comments from all reviewers): (1) added monthly time series information on each watershed (in addition of the annual time series, Supplement S6); (2) added a flowchart on the overall methodology; (3) the assumption on negligible inter-watershed groundwater flow (IGF) was clearly stated and discussed within the abstract, methods, discussion and conclusions sections; (4) added Appendix F showing a case study of imbalanced watersheds where IGF may not be the cause of UoP; and (5) tables on watershed characteristics were added to the manuscript (Table 1) and Supplement (Table S6).

GENERAL COMMENTS:

I would suggest the author to include the analysis for the summer monsoon season (i.e., June-July-August-September), as the summer monsoon accounts for a major portion of the annual precipitation over India.

Agree with the comment. Since the June-September time period typically has the largest contribution to rainfall and streamflow, time series graphics of these have been added for all of the stations (Supplement S6) where monthly streamflow observations were available (Peninsular India). Moreover, additional examples illustrating the monthly and annual water imbalance scenarios of $R > P_{obs}$ were included in the newly created Appendix F. These examples show that the imbalance is occurring not only at the annual scale but also at the monthly and seasonal time scale.

OTHER COMMENTS:

1. I would recommend the authors to use 'precipitation' instead of 'P', particularly in the introduction.

The word 'precipitation' was used instead of 'P' in the introduction. However, to be consistent with the rest of the manuscript, the use of 'P' for precipitation has been left unchanged in the remainder of the manuscript.

2. Line 34: "Raw data from P gauges" should be revised to "Raw data from rain gauges."

The phrase 'Raw data from P gauges' was revised to 'Raw data from rain gauges'.

3. Line 43: "However, gauge-based gridded datasets can be far from ideal" Do you have any evidence to support this statement for IMD data (concerned about word 'far')?"

The text has been changed to "However, gauge-based gridded datasets can suffer from inadequate representation of extreme events - such as those reported by King et al. (2013) in Australia; spurious trends due to changes in the locations of reporting gauges - such as those reported by Lin and Huybers (2019) using the IMD dataset; or uncertainties introduced by the relative positioning of reporting gauges - such as those reported by Prakash et al. (2019) using the IMD dataset".

4. Line 44: This paper, King et al. (2013), did not utilize the IMD data

Yes, King et al. (2013) did not utilize IMD data. The text has been revised to "However, gauge-based gridded datasets can suffer from inadequate representation of extreme events - such as those reported by King et al. (2013) in Australia;"

5. Line 46: What do you mean by 'other errors'? demonstrate them?"

The phrase "and other errors" was deleted since a discussion on these other errors is not directly relevant to this study.

6. Fig. 1: I would suggest plotting the rainfall estimates from the actual rain gauge stations along with the IMD gridded precipitation data. It could be possible (potentially) that the interpolation method used in constructing the IMD data introduces some biases.

It is possible that the grid resolution adopted by IMD (25 km) and the interpolation procedure used by IMD could introduce biases. However, station data is needed to investigate if biases are the result of interpolation procedures. Unfortunately, station data used by IMD is not available to the authors. Moreover, the goal of this study is not to compare IMD gridded product with station-level precipitation data – rather, the goal is to estimate gross underestimation of precipitation within IMD and other gridded precipitation data sources often used by the scientific community.

7. Fig. 1(b): I am concerned about the PBCOR data that authors have used to show the ratio of bias correction. The number of rain gauge stations used in this data (Fig. 2; Beck et al. 2020), particularly over India, is far fewer than the IMD rain gauge stations (Fig. 1; Prakash et al. 2015). So, it is possible that the observed largest ratios could be attributed to PBCOR datasets rather than IMD. I would suggest the authors to compare the PBCOR data with the IMD for both mean and extreme (e.g.; 99[th] percentile) cases before using it for bias correction ratio calculations.

Prakash, S., Mitra, A. K., Momin, I. M., Pai, D. S., Rajagopal, E. N., & Basu, S. (2015). Comparison of TMPA-3B42 versions 6 and 7 precipitation products with gauge-based data over India for the southwest monsoon period. Journal of Hydrometeorology, 16(1), 346-362.

We understand the reviewer's suggestion on making further comparisons between PBCOR and IMD – using metrics other than the long-term mean. The PBCOR dataset is a climatology and provides only long-term average values of annual and monthly precipitation. There is no daily data available. As

such, the suggested comparison cannot be made. We would also like to add that the PBCOR dataset by Beck et al 2020 was just one of the motivations behind this study. In this study the PBCOR dataset was used only to illustrate the extent of potential underestimation of precipitation by an independent study. As per the reviewer's suggestion, we included the paper by Prakash et al. (2015) in the discussion (Section 5.1.1, "Limitations with data").

8. Line 61: "If estimates from PBCOR are reasonable", follow comment 8

It appears that the reviewer meant to say 'follow comment 7' (instead of 'follow comment 8') since this is comment #8. As indicated in comment #7, the PBCOR dataset is a climatology and does not contain daily data. The PBCOR study estimated correction factors using a watershed imbalance analysis. We wanted to know if the correction factors from our study were consistent with PBCOR or not. Hence, PBCOR was brought into the discussion. The results of our study are completely independent of PBCOR.

9. Lines 93-95: "using procedures… watersheds of Northern India", How did you compile? Please explain in detail.

The following text has been added - "The catchment boundaries for the Northern Indian watersheds were derived using the HydroSHEDS suite of products, using the same procedures as the GHI dataset. Station descriptions available from CWC were validated using online maps (e.g., Google Maps). Stations were then relocated to the closest point on the river network. The watershed draining into this relocated point, and all of the upstream watersheds were recursively identified using a GIS software. Catchment areas for the delineated watersheds were validated against those reported by CWC."

10. Lines 97-100: Include discussion about uncertainties associated with the streamflow data

The uncertainties associated with streamflow measurements are briefly discussed in Section 5.1.1 ('Limitations with Data'). As per the reviewer's suggestion, additional literature on this topic has been added to the manuscript.

---

## Author Comment (AC2)

**Response to Reviewer #2's comments:**

The authors would like to thank the reviewer for the feedback and the suggestions made to improve the manuscript. The authors generally agree with both the major and minor comments, and revisions were made accordingly. Please see below for the specific responses to the individual comments.

**Summary of major changes** (based on comments from all reviewers): (1) added a flowchart on the overall methodology; (2) the assumption on negligible inter-watershed groundwater flow (IGF) was clearly stated and discussed within the abstract, methods, discussion and conclusions sections; (3) added Appendix F showing a case study of imbalanced watersheds where IGF may not be the cause of UoP; (4) added monthly time series information on each watershed (in addition of the annual time series, Supplement S6); and (5) tables on watershed characteristics were added to the manuscript (Table 1) and Supplement (Table S6).

**The major issues with the manuscript are:**

1. The study fails to account for the fundamental assumptions of the catchment water balance method. The primary assumptions of catchment/watershed water balance are of a closed system and overlapping physical and hydrological boundaries, i.e. the watershed boundaries overlap with their respective aquifer boundaries. Both rarely occur in nature. Most aquifers are leaky, and multiple watersheds may fall in a single aquifer, or adjacent watersheds may share a common aquifer. Thus catchments with leaky aquifers (https://wires.onlinelibrary.wiley.com/doi/abs/10.1002/wat2.1386 ) or non-overlapping aquifer-watershed boundaries with significant inter-basin groundwater transfer (https://www.sciencedirect.com/science/article/abs/pii/S0022169420300433), can have significant water imbalance, apart from the limitations mentioned in Section 5 of the manuscript. It may also explain why most off-balance catchments were seen in mountains, as they have significantly high geological connectivity.

We thank the reviewer for bringing up this important caveat which we did not address in the original manuscript. We understand the comment by the reviewer that water balance methods often assume that watersheds are self-contained units and water flowing out of a watershed has been generated within the watershed. The manuscript was revised at several places (including the abstract, methods, discussion and conclusions) to make the reader aware of this important assumption. The literature cited by the reviewer and other relevant references were added to the manuscript.

We acknowledge that some of the observed water imbalance could be due to inter-watershed groundwater flow (IGF). In the absence of field data on the geologic makeup of the watersheds and their groundwater flow patterns it is not possible to definitively know the effect of IGF on streamflow. However, we present examples (in the newly created Appendix F) to show that IGF is unlikely the cause of the observed water imbalance in certain watersheds.

The objective of this paper is not to oversimplify hydrological process representation, but to bring forth the apparent widespread and persistent issue of precipitation underestimation. Towards this goal we are trying to use available information to the extent feasible and are making reasonable assumptions

when needed. We acknowledge the limitations of both the information used and the assumptions made within our analysis.

2. The assumption that mountain and/or forested areas have minimal watershed management is not true. Many of the mountain regions and/or forested areas are included in watershed management through extensive government and civil society efforts. The activities include soil and water conservation efforts, which can change local and landscape-level hydrology. I would recommend the authors to have a relook at the assumption and its impact on the results.
https://www.cabidigitallibrary.org/doi/full/10.5555/20123144409 (Chauhan, 2010)

The phrase "minimal watershed management" needs additional clarification in the manuscript since it appears to give the reader the incorrect notion that there is hardly any management. The data on water management from India's water agencies (discussed in the manuscript) shows that for many watersheds within the mountainous and forested regions of India, the combined annual effect of management – such as surface water diversions (imports and exports), groundwater extraction, and reservoir are storage, is typically less than 20% of annual precipitation. This implies that for a watershed with an annual runoff coefficient of 0.4, the effect of management on streamflow = 20% / 0.4 = 50%. In other words, up to 50% of the annual streamflow could be affected by "minimal watershed management". Thus, "minimal watershed management" could still have a substantial effect on the annual streamflow.

We agree with the comment that watershed management may not be minimal in an absolute sense in mountainous or forested regions. It is important to note that we are making the above assumption at the watershed scale on an annual basis, and not at the landscape scale on a storm-event basis. The study suggested by the reviewer, Chauhan (2010), was added to the discussion (Section 5.1.1). Additional text has been added to the manuscript to clarify what is meant by "minimal watershed management". Moreover, within the limitations section we discuss the issues with the water management data used in the study.

Our objective is to find plausible explanations for the observed spuriously high annual runoff coefficients. Towards this objective we use heuristics to differentiate between spurious and non-spurious instances of the annual water balance. The choice of heuristics can affect the results of this study. Hence the sensitivity analyses on such heuristics in Section 4.1 ('Imbalanced watersheds using IMD-APHRO').

3. The majority of the watersheds are in central and southern India, which reduces the studies' representation.

Agree with the comment. Of the 242 watersheds analyzed in this study, 213 are in Peninsular India while 29 are in Northern India. Unfortunately, these are the only watersheds where streamflow data is publicly available. While data is collected by CWC (and other Indian water agencies) at gauging stations throughout India, such data is publicly available only for stations of Peninsular India. Data for the remainder of India is "classified" by these agencies and is unavailable to the authors. The data for Northern Indian watersheds had to be compiled from whatever limited information was available via reports published by water agencies.

This study demonstrates that even with a limited set of streamflow observations, one could identify that approximately a fourth to a third of the total watersheds in both Northern and Peninsular India are affected by water imbalance. It is hoped that this study would help bring forth the needed streamflow and hydrometeorological data.

4. I was excited to see a study on precipitation underestimation/overestimation. Few studies have approached doing hydrology backwards (https://agupubs.onlinelibrary.wiley.com/doi/10.1029/2008WR006912), and rarely any from India. However, the study loses a chance to dwell on the processes behind UoP, which could have been explored by going deeper in a selected watershed like Nethravati. The study shows an overall UoP scenario in central and southern India but doesn't inform us enough on why it might be happening, apart from data issues.

Following the reviewer's suggestion, the monthly and annual water balance of the Nethravati watershed (upstream of Bantwal station), and eight other adjacent watersheds in Southwestern India were further analyzed in the newly created Appendix F. These watersheds, which form the headwaters of several river basins, were shown to simultaneously experience monthly runoff coefficients of greater than1.0 (i.e., $R > P_{obs}$). If inter-watershed groundwater flow played a substantial part in the monthly (or annual) hydrology, some of the watersheds would experience higher runoff coefficients at the expense of the other watersheds. But that is not the case. Considering all other potential causes, including water management and terrestrial water storage changes, UoP appears to be the likely cause of water budget imbalance.

The authors realize that identification of gross UoP using a "reverse hydrology" approach (or based on the water balance approach used here) is a challenging task considering the limitations of available data and the number of proxy datasets that needed to be compiled. Field-scale data on the geological formations underlying each aquifer and the groundwater flow patterns within such formations is not available. In the absence of such data, it is not feasible to delve into the specific mechanisms behind each watershed's annual water imbalance. The authors also acknowledge that UoP may not be the cause of the observed annual water imbalance in some of the watersheds. It is hoped that the scientific community gets access to the needed data and other resources to help understand this very important issue of watershed imbalance and UoP.

**Detailed comments**

L100 Could you please add a table with salient features of the 242 watersheds. Size distribution, Elevation bands, rainfall characteristics, % forests, %ET, etc.

Yes. A table showing the characteristics of all 242 stations used in this study is added to the supplement (Table S6). The metrics within Table S6 include catchment area, elevation (max and median), percent area covered by forests and cropland, annual average precipitation and annual average ET. A new table showing the number of stations within each major basin and other relevant information on streamflow data is added as Table 1.

There seems to be a lack of watersheds from Himalaya, Western Ghats, Western and Central India. Can the study claim to represent India in the title when most of the watersheds are in central and southern India?

Similar to major comment #3, the watersheds used here are only those where observed streamflow data is publicly available. India's water agencies have additional streamflow data but is not available to the authors. Despite the limited availability of data, imbalanced watersheds occur in almost all of the major basins – particularly in the head waters of these major basins. By shedding light on the limitations of data availability this study hopes to help bring forth the needed data.

L118-119: What is the strength of interpolation between APHRODITE and IMD. Would recommend adding any results to supplementary table.

In this study the gauge-based APHRODITE dataset was used in portions of the study domain where IMD data was unavailable. The similarity between the two datasets has not been explored in this study. However, studies in the literature have compared the two datasets (e.g., Prakash et al. 2015) and found that APHRODITE compares reasonably to IMD across many parts of India. Limitations with APHRODITE have also been discussed in such studies. Based on the literature studies APHRODITE is assumed to be the best gauge-based alternative to IMD. The text in Section 2.2 was revised accordingly.

L149-150: The CGWB groundwater dataset is known to be quite limited by the number of wells and their representativeness. How does that effect the manuscript?

Yes, the CGWB dataset is limited by the number of wells and the recharge estimates from CGWB are dependent on such data and the many assumptions made within their analyses. Some studies have discussed the pitfalls with analyzing trends in groundwater levels using the CGWB dataset (e.g., Hora et al. 2020). However, the CGWB dataset is the only observation-based dataset on groundwater extraction available for the entire study domain. Additional text has been added in Section 5.1.1 ('Limitations with Data').

L162-165: Are there any studies on the validation of TWS from GRACE against ground data? Please do add a section on the limitations of GRACE, especially in complex mountains or geologically diverse systems.

Some of the limitations of GRACE data have been discussed in Section 5.1.1 ("Limitations with data"). The effective resolution of GRACE data is about 300 km x 300 km. As such, GRACE captures only large-scale changes and cannot be directly compared with point-scale (e.g., groundwater wells) or small-scale changes in water table depth. Additional text has been added in Section 5.1.1 ('Limitations with Data').

L175 There seems to be an objective-method mismatch. The methods section describes the theoretical and per-watershed aspects well, but a flowchart mimicking the methods followed to achieve the three objectives would help us understand the study.

Agree with the comment. In order to help the reader better understand the objectives of this study and the methods, a flow chart on the overall methodology was added as Figure 1 in Section 1. This new figure complements Figure 7 (Figure 6, original manuscript; schematic on identification of spurious scenarios) and helps the reader better understand the overall analysis.

L175-180 What about leaky aquifers and inter-connected basins with significant groundwater transfer?

Similar to major comment #1, it is not known to what extent the watersheds analyzed in this study are affected by inter-watershed groundwater flow (IGF) – either because of 'gaining' aquifers or 'losing' (or leaky) aquifers. In the absence of field data on groundwater flow patterns within each watershed it is not possible to quantify the effect of IGF on streamflow. However, we present examples in the newly created Appendix F to show that IGF is not likely the explanation for the observed water imbalance in certain watersheds. Only field-level data can help establish the actual cause of watershed imbalance.

L200-205 Could you please define the exports and imports as per CGWB 2019 and how they are distinct from ∆GW natrual, ∆GW human and ∆Reservoir, with examples if possible?

Yes. Exports and imports are net surface water diversions and represent the net loss of water and net gain of water, respectively. ∆GW natural and ∆GW human are natural and artificial changes to groundwater, respectively. ∆Reservoir is the change in reservoir storage. Text has been added in Section 3.1 to clarify these definitions.

L210-211 The assumption is usually appropriate in catchments with high infiltration and transmissivity, eg Karst aquifers. Do any of the watersheds have such physical conditions?

Since this study does not assume that net change in annual storage is negligible, we have not investigated the watersheds where such change is negligible. However, we identified relevant literature on Karst aquifers (e.g., Dar et al. 2014), and included it in the manuscript (Section 5.1.2, 'Limitations with the methodology').

L240 – 245 This is a big assumption to make. Many of the mountain regions and/or forested areas are included in watershed management through extensive government and civil society efforts. The activities include soil and water conservation efforts which can change local and landscape level hydrology. Please have a relook. https://www.cabidigitallibrary.org/doi/full/10.5555/20123144409

Similar to major comment #2, we agree with the comment that mountainous or forested regions cannot be assumed to have minimal watershed management in an absolute sense of the word "minimal". Additional text was added to the manuscript (Section 3.1) to help the reader understand that "minimal" is used in a relative sense, and one could still have a substantial portion of the annual streamflow affected by "minimal management".

L246 Is off-balance the same as imbalance here? Please maintain consistency if they are the same or clarify if not.

Yes, "off-balance" and "imbalance" were used interchangeably in the original manuscript. As per the reviewer's suggestion, the word "off-balance" was replaced with "imbalance" throughout the revised manuscript and the revised supplement.

L258-260 Please provide details of the spatial syncing methods and any sensitivity analysis to check if there was any loss of information.

In order to compare streamflow at the outlet of the catchment with grid-based precipitation (ET, etc.) over the entire catchment, one has to aggregate the gridded datasets to the domain of the catchment. An area-weighted scheme which accounted for the spatial overlap of the catchment boundaries and individual grid cells was used to perform the aggregation. The GIS procedure employed here is typical

to hydrologic studies. There is no loss of information within this aggregation process. The GIS procedure is illustrated through the schematic in Figure 8 within Section 3.2 (Figure 7, original manuscript).

---

## Referee Report (RR1)

[referee-annotated manuscript omitted]

---

## Author Response (AR4)

**Summary of Editor's Comments & Responses: Iteration 3**

**Response to Editor's comments:**

Thank you for providing your responses to the two reviewers' comments. Based on your responses, I understand your reasons for not conducting further analysis as suggested by the reviewer. However, some text adjustments must be made before the paper can be accepted for publication in HESS. I found the text rather lengthy, with many repetitions, and not very concise. Detailed comments and suggestions regarding how the text should be revised can be found in the attached document. Please do not feel obligated to agree with all my comments, but the message is that the text should be more focused without losing the scientific content.

The authors would like to thank the editor for the comments. We agree with the general comment that the text is sometimes unnecessarily long and can be shortened. Following the comments, we moved the less important information to the Supplement, reduced the size of some of the large graphics, and condensed the text where possible. These changes were made ensuring the essence of the paper remained the same and important details were not discarded. The manuscript is now 34 pages long compared to the original 47 pages.

Please see below the specific responses to the major comments. Minor revisions such as text revisions and deletions suggested by the editor were made throughout the manuscript but not listed here.

Major Comments:

Line 1: I found the abstract to be a bit too long. Please shorten it to include only the motivation, what was done, and key results. It is now too detailed.

The abstract was shortened from 31 lines to 21 lines (little over half a page).

Line 64: Please move all appendixes to the supporting material (SM), i.e., uploaded to the HESS system as a separate file so it will not be part of the main text.

In the prior version, the Appendices described some of the data and the supplementary material (SM) had graphics and additional material (data, GIS files, etc.). In the revised version, all appendices are part of the SM. The entire SM document was revised accordingly and will be uploaded as a separate file.

Line 65: Move to the SM. Shouldn't be presented as part of the main text.

Description of the PBCOR dataset and graphics were moved to the SM.

Line 80: Move to the method section.

Introduction has been revised and the text moved to the Methods section.

Line 89: In data or methods - but this is not related to the introduction

Introduction has been revised and the text moved to the Data section.

Line 108: Change all "Figure X (panel(y))" to "Fig. Xy".

The above change was made everywhere, except the beginning of the sentence where HESS journal template requires the use of the word "Figure".

Line 112: Where is the reference to Table 1 in the text? Also - could this be presented as SM?

This table was moved to the SM.

Line 123: The paper is a bit lengthly and in many places it can be shorten. This is a good example. This information is not needed.

The unnecessary text was deleted as suggested. Changes were made throughout the manuscript as per this suggestion.

Line 148: Add the relevant labels in the figure

The figure was reduced in size but otherwise left unchanged for the sake of convenience. The description of the text was condensed to about half its original size.

Line 151: Can be considerably shorten. This entire paragraph can be summarized in two sentences, pointing on the land use product that was used and indicating the uses that were grouped together. No need to inform the readers the spatial allocation of the units - it is clearly visible in the figure.

The description was condensed to about half its original size.

Line 157: I suggest merging the two panels into one (for example, contours for the elevation and colors for the land cover) so the figure can be a single column (8 cm width) instead of two columns.

The color scheme for elevation is a bit awkward. Why not following a traditional color scheme, e.g., changing from blue (low) to red (high)?

The figure was reduced in size but otherwise left unchanged for the sake of convenience. We agree that the color scheme is somewhat unusual. Since a number of graphics within the paper and the SM document use a similar color scheme, we left the figure unchanged.

Line 244 (Figure 7): I don't see the need for this figure to stretch over an entire page. It is okay to keep the width stretched over two-columns (i.e., 16 cm), but please reduce it to half page maximum, or even less (i.e., up to 12 cm). No need to repeat Eq. 5 for example.

The figure was reduced from its original size to about half a page.

Line 244: You are often using the term "in this study" or similar expressions. I think it is clear that everything that is presented in the paper is relevant to "this study". I would remove most of these indications (I counted more than 20 so far...).

The phrase "in this study" was used too often. It was removed wherever possible. In the revised manuscript, there are only three instances of them.

Line 287: Present as SM

This figure was moved to the SM.

Line 422: Figure 12: Please see if you can rescale this figure to be a single column (8 cm width) figure.

This figure has been rescaled to a single column figure.

Line 423: I am not sure this summary is needed. If you prefer to keep it, maybe you can think how to shorten it into a single paragraph mentioning only the key results you would like to highlight?

This summary has been deleted.

Line 634: Appendices: Change appendixes to SM and show in a separate file.

All appendices are part of the SM. The SM document has been revised accordingly.

**Summary of Reviewer Comments & Responses: Iteration 2**

**Response to Referee #1's comments:**

I appreciate the great efforts the authors have made in response to my questions and concerns. The revision clarifies nearly all the points I raised. I would suggest authors to include the suggestions provided below in the paper before it is considered for publication.

Minor comments:

Lines 40-45: "Within these studies, gauge-based precipitation products are often treated as reference products, or benchmarks, when evaluating satellite-based and other non-traditional datasets." Here, it would be nice to provide some examples of non-traditional datasets, along with citations. I would suggest citing the relevant research articles provided below.

1. Shahi, N. K. (2022). Fidelity of the latest high-resolution CORDEX-CORE regional climate model simulations in the representation of the Indian summer monsoon precipitation characteristics. Climate Dynamics, 1-23. https://doi.org/10.1007/s00382-022-06602-9.

2. Shahi, N. K., Rai, S., Sahai, A. K., & Abhilash, S. (2018). Intra-seasonal variability of the South Asian monsoon and its relationship with the Indo–Pacific sea-surface temperature in the NCEP CFSv2. International Journal of Climatology, 38, e28-e47. https://doi.org/10.1002/joc.5349.

The first reference also supports your sentence provided in Lines 45-50 "However,gauge-based gridded datasets can suffer from inadequate representation of extreme event..."

The authors would like to thank the reviewer for the feedback on the revised manuscript. We agree with the reviewer that the first article mentioned above is relevant to the general discussion on precipitation datasets across India. The above article also analyzed some of the datasets used in this manuscript. This article will be added to the text (in Section 1) and will be included in the citations. We do not think the second article mentioned above is directly relevant to the manuscript and so we did not include it in the citations.

**Response to Referee #2's comments:**

The authors would like to thank the reviewer for the feedback on the revised manuscript.

In order to understand the effect of interbasin groundwater flow (IGF) on the water budget, the reviewer is suggesting case studies of watersheds (and their aquifers) which were identified to be affected by precipitation underestimation – the suggested work by the reviewer includes extensive literature review, data compilation and data analysis. The goal of this study is to identify potential underestimation of precipitation across the whole study domain, and not just a specific region. While regional and local studies are valuable, one would require extensive data as well as regional/local expertise to perform such studies. It is beyond the scope of this study to perform such case studies. We hope that our study encourages water agencies to share valuable data on precipitation, streamflow and other hydrological variables, and motivates the community to better quantify hydrologic budgets across India's watersheds.

Please see below the specific responses to the individual comments.

Dear Editor,
Thank you for considering me as a reviewer of the manuscript. The authors have attempted to address the comments and suggestions given in the first review, which has improved the manuscript. However, there still remain a few major concerns and areas of improvement:

1. The issue of IGF is acknowledged and addressed by a case study approach where adjoining watersheds in Southern India are studied for UoP. Authors rule out IGF by observing simultaneous occurrences of >1 runoff ratios in high-elevation watersheds. However, in most regions, aquifers occur at multiple levels, and I recommend Authors check the NAQUIM project and the data available with them for these and surrounding watersheds (https://www.aims-cgwb.org/index.php). The aquifer mapping in the region suggests the presence of two aquifers at different depths, which may potentially contribute to neighbouring watersheds despite elevational differences.
While I understand the limitations in accessing precise aquifer maps for India, the assumption of negligible IGF Is not justified as a physical reality. Assuming otherwise due to a lack of data would distort the interpretations of the physical system and precipitation hydrology.

In the example discussed in Appendix F, the reviewer is asking us to investigate if groundwater from lower-elevation aquifers can contribute to groundwater at higher-elevation aquifers, using the data sources outlined above. While a preliminary review of the above mentioned data sources indicates the presence of some relevant information, we believe that it is not possible to adequately address the reviewer's concerns within the scope of this study. We do not have the resources nor the expertise to pursue such an investigation. We would like to emphasize that we do not assume interbasin groundwater flow (IGF) to be negligible or rule out its existence. We merely provided an example (Appendix F) where IGF is unlikely the cause of the observed watershed imbalance.

A case study approach is suitable in such cases to exemplify the proof of concept and I recommend authors use District Resource Maps (DRMs) by the Geological Society of India, the aquifer management plans by NAQUIM project, and secondary literature to develop an improved understanding of regional aquifers and the presence/absence of significant IGF in select watersheds with the observed UoP.

The suggested case studies to understand and quantify IGF require extensive field-scale data along with regional/local expertise. While the above mentioned data sources have some relevant information, we believe that it is not possible to adequately address the reviewer's concerns within the scope of this study.

I would also recommend incorporating the case study into the main manuscript body and discussing the assumptions, hypothesis, and results accordingly with appropriate citations.

As discussed in the manuscript, observed imbalance could be due to several factors, viz., underestimation of precipitation, IGF, water management, terrestrial water storage changes or other factors not considered in this study. Since the relative importance of IGF is unknown, we do not want to overemphasize it. Hence, we chose not to move the case study (Appendix F) to the main part of the manuscript.

2. In the absence of India-level data on most parameters, as acknowledged by the authors, perhaps it would be better to focus on developing representative case studies wherever most information is available.
It would be pertinent to find as much secondary literature as possible for at least a subset of the study area. I would recommend authors do an exhaustive watershed-level review of studies on water balance in at least a representation set of the "imbalanced" watersheds. It would help contextualise the findings from a triangulation approach, i.e., ET+streamflow records, aquifer properties (DRM/NAQUIM), and water management (secondary literature).

The reviewer is suggesting an exhaustive literature review on imbalanced watersheds followed by a "triangulation" approach. We are unsure if we would be able to address the reviewer's concerns after such a review. For the suggested "triangulation" approach, we neither have reliable ET data (as discussed in the manuscript) nor have adequate information on aquifer flow regimes and local-scale water management. It is beyond the scope of this study to perform the above suggested literature review and analyses.

3. The study uses IMD-APHRODITE blended product in IMD-missing regions. Both gauge-based products are known to diverge in certain landscapes and seasons as per recent studies (Prakash et al. (2015) is one of the earliest comparisons). I recommend authors perform a comparative assessment of the two products, a pre-requisite to grided product blending, and report the findings in the annexure. Another simple check would be to conduct a buffered pixel-wise comparison for overlapping regions (both IMD & APHRODITE) around the IMD-missing regions and report the bias, if any, and its impact on the results.

For those regions of the study domain where IMD is unavailable (outside of India's boundaries), we used the gauge-based APHRODITE dataset to supplement IMD. Instead of assuming zero precipitation for regions outside of India, we are using the data from APHRODITE. The reasons for choosing APHRODITE were discussed in Section 2.2 of the manuscript. Also, we included text on studies comparing IMD with APHRODITE. Unless adequate precipitation gauge data is available, it is not possible to assess the differences between these datasets and their relative merits and demerits. Identifying the differences between precipitation datasets for specific regions and seasons is beyond the scope of this study. Moreover, the goal of this study is to identify potential underestimation of precipitation and is not an inter-comparison of precipitation datasets.

I think the manuscript addresses a pertinent issue and recommend the manuscript for a major revision again to address the abovementioned conceptual issues.

**Summary of Reviewer Comments & Responses: Iteration 1**

**Response to Reviewer #1's comments:**

The manuscript entitled "Extent of gross underestimation of precipitation in India" by Goteti and Famiglietti, analyzes watershed-scale underestimation of precipitation over India using various precipitation datasets within a water imbalance analysis. The authors have designed the study with relevant methodology through the knowledge attained from associated literature. There are some corrections required in the manuscript, although the authors have explained the results and the methodology in detail, there are a few disconnections between sentences and minor grammar corrections required. I have raised a few questions that require clarification, also suggested some necessary modifications and additions to the manuscript.

The authors would like to thank the reviewer for the feedback and the suggestions made to improve the manuscript. The authors generally agree with the comments, and revisions were made accordingly. Please see below the specific responses to the individual comments.

**Summary of major changes** (based on comments from all reviewers): (1) added monthly time series information on each watershed (in addition of the annual time series, Supplement S6); (2) added a flowchart on the overall methodology; (3) the assumption on negligible inter-watershed groundwater flow (IGF) was clearly stated and discussed within the abstract, methods, discussion and conclusions sections; (4) added Appendix F showing a case study of imbalanced watersheds where IGF may not be the cause of UoP; and (5) tables on watershed characteristics were added to the manuscript (Table 1) and Supplement (Table S6).

GENERAL COMMENTS:

I would suggest the author to include the analysis for the summer monsoon season (i.e., June-July-August-September), as the summer monsoon accounts for a major portion of the annual precipitation over India.

Agree with the comment. Since the June-September time period typically has the largest contribution to rainfall and streamflow, time series graphics of these have been added for all of the stations (Supplement S6) where monthly streamflow observations were available (Peninsular India). Moreover, additional examples illustrating the monthly and annual water imbalance scenarios of $R > P_{obs}$ were included in the newly created Appendix F. These examples show that the imbalance is occurring not only at the annual scale but also at the monthly and seasonal time scale.

OTHER COMMENTS:

1. I would recommend the authors to use 'precipitation' instead of 'P', particularly in the introduction.

The word 'precipitation' was used instead of 'P' in the introduction. However, to be consistent with the rest of the manuscript, the use of 'P' for precipitation has been left unchanged in the remainder of the manuscript.

2. Line 34: "Raw data from P gauges" should be revised to "Raw data from rain gauges."

The phrase 'Raw data from P gauges' was revised to 'Raw data from rain gauges'.

3. Line 43: "However, gauge-based gridded datasets can be far from ideal" Do you have any evidence to support this statement for IMD data (concerned about word 'far')?"

The text has been changed to "However, gauge-based gridded datasets can suffer from inadequate representation of extreme events - such as those reported by King et al. (2013) in Australia; spurious trends due to changes in the locations of reporting gauges - such as those reported by Lin and Huybers (2019) using the IMD dataset; or uncertainties introduced by the relative positioning of reporting gauges - such as those reported by Prakash et al. (2019) using the IMD dataset".

4. Line 44: This paper, King et al. (2013), did not utilize the IMD data

Yes, King et al. (2013) did not utilize IMD data. The text has been revised to "However, gauge-based gridded datasets can suffer from inadequate representation of extreme events - such as those reported by King et al. (2013) in Australia;"

5. Line 46: What do you mean by 'other errors'? demonstrate them?"

The phrase "and other errors" was deleted since a discussion on these other errors is not directly relevant to this study.

6. Fig. 1: I would suggest plotting the rainfall estimates from the actual rain gauge stations along with the IMD gridded precipitation data. It could be possible (potentially) that the interpolation method used in constructing the IMD data introduces some biases.

It is possible that the grid resolution adopted by IMD (25 km) and the interpolation procedure used by IMD could introduce biases. However, station data is needed to investigate if biases are the result of interpolation procedures. Unfortunately, station data used by IMD is not available to the authors. Moreover, the goal of this study is not to compare IMD gridded product with station-level precipitation data – rather, the goal is to estimate gross underestimation of precipitation within IMD and other gridded precipitation data sources often used by the scientific community.

7. Fig. 1(b): I am concerned about the PBCOR data that authors have used to show the ratio of bias correction. The number of rain gauge stations used in this data (Fig. 2; Beck et al. 2020), particularly over India, is far fewer than the IMD rain gauge stations (Fig. 1; Prakash et al. 2015). So, it is possible that the observed largest ratios could be attributed to PBCOR datasets rather than IMD. I would suggest the authors to compare the PBCOR data with the IMD for both mean and extreme (e.g.; 99[th] percentile) cases before using it for bias correction ratio calculations.

Prakash, S., Mitra, A. K., Momin, I. M., Pai, D. S., Rajagopal, E. N., & Basu, S. (2015). Comparison of TMPA-3B42 versions 6 and 7 precipitation products with gauge-based data over India for the southwest monsoon period. Journal of Hydrometeorology, 16(1), 346-362.

We understand the reviewer's suggestion on making further comparisons between PBCOR and IMD – using metrics other than the long-term mean. The PBCOR dataset is a climatology and provides only long-term average values of annual and monthly precipitation. There is no daily data available. As such, the suggested comparison cannot be made. We would also like to add that the PBCOR dataset by Beck et al 2020 was just one of the motivations behind this study. In this study the PBCOR dataset was used only to illustrate the extent of potential underestimation of precipitation by an independent study. As per the reviewer's suggestion, we included the paper by Prakash et al. (2015) in the discussion (Section 5.1.1, "Limitations with data").

8. Line 61: "If estimates from PBCOR are reasonable", follow comment 8

It appears that the reviewer meant to say 'follow comment 7' (instead of 'follow comment 8') since this is comment #8. As indicated in comment #7, the PBCOR dataset is a climatology and does not contain daily data. The PBCOR study estimated correction factors using a watershed imbalance analysis. We wanted to know if the correction factors from our study were consistent with PBCOR or not. Hence, PBCOR was brought into the discussion. The results of our study are completely independent of PBCOR.

9. Lines 93-95: "using procedures… watersheds of Northern India", How did you compile? Please explain in detail.

The following text has been added - "The catchment boundaries for the Northern Indian watersheds were derived using the HydroSHEDS suite of products, using the same procedures as the GHI dataset. Station descriptions available from CWC were validated using online maps (e.g., Google Maps). Stations were then relocated to the closest point on the river network. The watershed draining into this relocated point, and all of the upstream watersheds were recursively identified using a GIS software. Catchment areas for the delineated watersheds were validated against those reported by CWC."

10. Lines 97-100: Include discussion about uncertainties associated with the streamflow data

The uncertainties associated with streamflow measurements are briefly discussed in Section 5.1.1 ('Limitations with Data'). As per the reviewer's suggestion, additional literature on this topic has been added to the manuscript.

**Response to Reviewer #2's comments:**

The authors would like to thank the reviewer for the feedback and the suggestions made to improve the manuscript. The authors generally agree with both the major and minor comments, and revisions were made accordingly. Please see below for the specific responses to the individual comments.

**Summary of major changes** (based on comments from all reviewers): (1) added a flowchart on the overall methodology; (2) the assumption on negligible inter-watershed groundwater flow (IGF) was clearly stated and discussed within the abstract, methods, discussion and conclusions sections; (3) added Appendix F showing a case study of imbalanced watersheds where IGF may not be the cause of UoP; (4) added monthly time series information on each watershed (in addition of the annual time series, Supplement S6); and (5) tables on watershed characteristics were added to the manuscript (Table 1) and Supplement (Table S6).

**The major issues with the manuscript are:**

1. The study fails to account for the fundamental assumptions of the catchment water balance method. The primary assumptions of catchment/watershed water balance are of a closed system and overlapping physical and hydrological boundaries, i.e. the watershed boundaries overlap with their respective aquifer boundaries. Both rarely occur in nature. Most aquifers are leaky, and multiple watersheds may fall in a single aquifer, or adjacent watersheds may share a common aquifer. Thus catchments with leaky aquifers (https://wires.onlinelibrary.wiley.com/doi/abs/10.1002/wat2.1386 ) or non-overlapping aquifer-watershed boundaries with significant inter-basin groundwater transfer (https://www.sciencedirect.com/science/article/abs/pii/S0022169420300433), can have significant water imbalance, apart from the limitations mentioned in Section 5 of the manuscript. It may also explain why most off-balance catchments were seen in mountains, as they have significantly high geological connectivity.

We thank the reviewer for bringing up this important caveat which we did not address in the original manuscript. We understand the comment by the reviewer that water balance methods often assume that watersheds are self-contained units and water flowing out of a watershed has been generated within the watershed. The manuscript was revised at several places (including the abstract, methods, discussion and conclusions) to make the reader aware of this important assumption. The literature cited by the reviewer and other relevant references were added to the manuscript.

We acknowledge that some of the observed water imbalance could be due to inter-watershed groundwater flow (IGF). In the absence of field data on the geologic makeup of the watersheds and their groundwater flow patterns it is not possible to definitively know the effect of IGF on streamflow. However, we present examples (in the newly created Appendix F) to show that IGF is unlikely the cause of the observed water imbalance in certain watersheds.

The objective of this paper is not to oversimplify hydrological process representation, but to bring forth the apparent widespread and persistent issue of precipitation underestimation. Towards this goal we

are trying to use available information to the extent feasible and are making reasonable assumptions when needed. We acknowledge the limitations of both the information used and the assumptions made within our analysis.

2. The assumption that mountain and/or forested areas have minimal watershed management is not true. Many of the mountain regions and/or forested areas are included in watershed management through extensive government and civil society efforts. The activities include soil and water conservation efforts, which can change local and landscape-level hydrology. I would recommend the authors to have a relook at the assumption and its impact on the results.
https://www.cabidigitallibrary.org/doi/full/10.5555/20123144409 (Chauhan, 2010)

The phrase "minimal watershed management" needs additional clarification in the manuscript since it appears to give the reader the incorrect notion that there is hardly any management. The data on water management from India's water agencies (discussed in the manuscript) shows that for many watersheds within the mountainous and forested regions of India, the combined annual effect of management – such as surface water diversions (imports and exports), groundwater extraction, and reservoir are storage, is typically less than 20% of annual precipitation. This implies that for a watershed with an annual runoff coefficient of 0.4, the effect of management on streamflow = 20% / 0.4 = 50%. In other words, up to 50% of the annual streamflow could be affected by "minimal watershed management". Thus, "minimal watershed management" could still have a substantial effect on the annual streamflow.

We agree with the comment that watershed management may not be minimal in an absolute sense in mountainous or forested regions. It is important to note that we are making the above assumption at the watershed scale on an annual basis, and not at the landscape scale on a storm-event basis. The study suggested by the reviewer, Chauhan (2010), was added to the discussion (Section 5.1.1). Additional text has been added to the manuscript to clarify what is meant by "minimal watershed management". Moreover, within the limitations section we discuss the issues with the water management data used in the study.

Our objective is to find plausible explanations for the observed spuriously high annual runoff coefficients. Towards this objective we use heuristics to differentiate between spurious and non-spurious instances of the annual water balance. The choice of heuristics can affect the results of this study. Hence the sensitivity analyses on such heuristics in Section 4.1 ('Imbalanced watersheds using IMD-APHRO').

3. The majority of the watersheds are in central and southern India, which reduces the studies' representation.

Agree with the comment. Of the 242 watersheds analyzed in this study, 213 are in Peninsular India while 29 are in Northern India. Unfortunately, these are the only watersheds where streamflow data is publicly available. While data is collected by CWC (and other Indian water agencies) at gauging stations throughout India, such data is publicly available only for stations of Peninsular India. Data for the remainder of India is "classified" by these agencies and is unavailable to the authors. The data for Northern Indian watersheds had to be compiled from whatever limited information was available via

reports published by water agencies.

This study demonstrates that even with a limited set of streamflow observations, one could identify that approximately a fourth to a third of the total watersheds in both Northern and Peninsular India are affected by water imbalance. It is hoped that this study would help bring forth the needed streamflow and hydrometeorological data.

4. I was excited to see a study on precipitation underestimation/overestimation. Few studies have approached doing hydrology backwards (https://agupubs.onlinelibrary.wiley.com/doi/10.1029/2008WR006912), and rarely any from India. However, the study loses a chance to dwell on the processes behind UoP, which could have been explored by going deeper in a selected watershed like Nethravati. The study shows an overall UoP scenario in central and southern India but doesn't inform us enough on why it might be happening, apart from data issues.

Following the reviewer's suggestion, the monthly and annual water balance of the Nethravati watershed (upstream of Bantwal station), and eight other adjacent watersheds in Southwestern India were further analyzed in the newly created Appendix F. These watersheds, which form the headwaters of several river basins, were shown to simultaneously experience monthly runoff coefficients of greater than1.0 (i.e., $R > P_{obs}$). If inter-watershed groundwater flow played a substantial part in the monthly (or annual) hydrology, some of the watersheds would experience higher runoff coefficients at the expense of the other watersheds. But that is not the case. Considering all other potential causes, including water management and terrestrial water storage changes, UoP appears to be the likely cause of water budget imbalance.

The authors realize that identification of gross UoP using a "reverse hydrology" approach (or based on the water balance approach used here) is a challenging task considering the limitations of available data and the number of proxy datasets that needed to be compiled. Field-scale data on the geological formations underlying each aquifer and the groundwater flow patterns within such formations is not available. In the absence of such data, it is not feasible to delve into the specific mechanisms behind each watershed's annual water imbalance. The authors also acknowledge that UoP may not be the cause of the observed annual water imbalance in some of the watersheds. It is hoped that the scientific community gets access to the needed data and other resources to help understand this very important issue of watershed imbalance and UoP.

**Detailed comments**

L100 Could you please add a table with salient features of the 242 watersheds. Size distribution, Elevation bands, rainfall characteristics, % forests, %ET, etc.

Yes. A table showing the characteristics of all 242 stations used in this study is added to the supplement (Table S6). The metrics within Table S6 include catchment area, elevation (max and median), percent area covered by forests and cropland, annual average precipitation and annual

average ET. A new table showing the number of stations within each major basin and other relevant information on streamflow data is added as Table 1.

There seems to be a lack of watersheds from Himalaya, Western Ghats, Western and Central India. Can the study claim to represent India in the title when most of the watersheds are in central and southern India?

Similar to major comment #3, the watersheds used here are only those where observed streamflow data is publicly available. India's water agencies have additional streamflow data but is not available to the authors. Despite the limited availability of data, imbalanced watersheds occur in almost all of the major basins – particularly in the head waters of these major basins. By shedding light on the limitations of data availability this study hopes to help bring forth the needed data.

L118-119: What is the strength of interpolation between APHRODITE and IMD. Would recommend adding any results to supplementary table.

In this study the gauge-based APHRODITE dataset was used in portions of the study domain where IMD data was unavailable. The similarity between the two datasets has not been explored in this study. However, studies in the literature have compared the two datasets (e.g., Prakash et al. 2015) and found that APHRODITE compares reasonably to IMD across many parts of India. Limitations with APHRODITE have also been discussed in such studies. Based on the literature studies APHRODITE is assumed to be the best gauge-based alternative to IMD. The text in Section 2.2 was revised accordingly.

L149-150: The CGWB groundwater dataset is known to be quite limited by the number of wells and their representativeness. How does that effect the manuscript?

Yes, the CGWB dataset is limited by the number of wells and the recharge estimates from CGWB are dependent on such data and the many assumptions made within their analyses. Some studies have discussed the pitfalls with analyzing trends in groundwater levels using the CGWB dataset (e.g., Hora et al. 2020). However, the CGWB dataset is the only observation-based dataset on groundwater extraction available for the entire study domain. Additional text has been added in Section 5.1.1 ('Limitations with Data').

L162-165: Are there any studies on the validation of TWS from GRACE against ground data? Please do add a section on the limitations of GRACE, especially in complex mountains or geologically diverse systems.

Some of the limitations of GRACE data have been discussed in Section 5.1.1 ("Limitations with data"). The effective resolution of GRACE data is about 300 km x 300 km. As such, GRACE captures only large-scale changes and cannot be directly compared with point-scale (e.g., groundwater wells) or small-scale changes in water table depth. Additional text has been added in Section 5.1.1 ('Limitations with Data').

L175 There seems to be an objective-method mismatch. The methods section describes the theoretical and per-watershed aspects well, but a flowchart mimicking the methods followed to achieve the three objectives would help us understand the study.

Agree with the comment. In order to help the reader better understand the objectives of this study and the methods, a flow chart on the overall methodology was added as Figure 1 in Section 1. This new figure complements Figure 7 (Figure 6, original manuscript; schematic on identification of spurious scenarios) and helps the reader better understand the overall analysis.

L175-180 What about leaky aquifers and inter-connected basins with significant groundwater transfer?

Similar to major comment #1, it is not known to what extent the watersheds analyzed in this study are affected by inter-watershed groundwater flow (IGF) – either because of 'gaining' aquifers or 'losing' (or leaky) aquifers. In the absence of field data on groundwater flow patterns within each watershed it is not possible to quantify the effect of IGF on streamflow. However, we present examples in the newly created Appendix F to show that IGF is not likely the explanation for the observed water imbalance in certain watersheds. Only field-level data can help establish the actual cause of watershed imbalance.

L200-205 Could you please define the exports and imports as per CGWB 2019 and how they are distinct from ∆GW natrual, ∆GW human and ∆Reservoir, with examples if possible?

Yes. Exports and imports are net surface water diversions and represent the net loss of water and net gain of water, respectively. ∆GW natural and ∆GW human are natural and artificial changes to groundwater, respectively. ∆Reservoir is the change in reservoir storage. Text has been added in Section 3.1 to clarify these definitions.

L210-211 The assumption is usually appropriate in catchments with high infiltration and transmissivity, eg Karst aquifers. Do any of the watersheds have such physical conditions?

Since this study does not assume that net change in annual storage is negligible, we have not investigated the watersheds where such change is negligible. However, we identified relevant literature on Karst aquifers (e.g., Dar et al. 2014), and included it in the manuscript (Section 5.1.2, 'Limitations with the methodology').

L240 – 245 This is a big assumption to make. Many of the mountain regions and/or forested areas are included in watershed management through extensive government and civil society efforts. The activities include soil and water conservation efforts which can change local and landscape level hydrology. Please have a relook. https://www.cabidigitallibrary.org/doi/full/10.5555/20123144409

Similar to major comment #2, we agree with the comment that mountainous or forested regions cannot be assumed to have minimal watershed management in an absolute sense of the word "minimal". Additional text was added to the manuscript (Section 3.1) to help the reader understand that "minimal" is used in a relative sense, and one could still have a substantial portion of the annual streamflow affected by "minimal management".

L246 Is off-balance the same as imbalance here? Please maintain consistency if they are the same or clarify if not.

Yes, "off-balance" and "imbalance" were used interchangeably in the original manuscript. As per the reviewer's suggestion, the word "off-balance" was replaced with "imbalance" throughout the revised manuscript and the revised supplement.

L258-260 Please provide details of the spatial syncing methods and any sensitivity analysis to check if there was any loss of information.

In order to compare streamflow at the outlet of the catchment with grid-based precipitation (ET, etc.) over the entire catchment, one has to aggregate the gridded datasets to the domain of the catchment. An area-weighted scheme which accounted for the spatial overlap of the catchment boundaries and individual grid cells was used to perform the aggregation. The GIS procedure employed here is typical to hydrologic studies. There is no loss of information within this aggregation process. The GIS procedure is illustrated through the schematic in Figure 8 within Section 3.2 (Figure 7, original manuscript).